# WHEN COVARIATE-SHIFTED DATA AUGMENTATION INCREASES TEST ERROR AND HOW TO FIX IT

## ABSTRACT

We study covariate-shifted data augmentation where the augmented targets are drawn from the true predictive distribution but the inputs are shifted. Empirically, some forms of data augmentation such as adversarial training improve robustness, but increase test error. We provide precise conditions under which data augmentation can increase test error for minimum norm interpolation estimators in linear regression. As a fix, we propose X-regularization which uses unlabeled data to regularize the parameters towards the non-augmented estimate. We prove that augmentation with X-regularization never increases test error in linear regression. Empirically, X-regularization consistently improves both robustness and standard test error across different adversarial training algorithms and perturbations on CIFAR-10.

## 1 INTRODUCTION

Adversarial training improves the robustness of neural networks to perturbations, commonly referred to as adversarial examples (Goodfellow et al., 2015; Szegedy et al., 2014; Biggio et al., 2013). However, adversarial training also causes an undesirable increase in the error on the unperturbed images (test error). How do we obtain both robust and accurate networks? At the core, adversarial training is a form of data augmentation where we augment the training set with worst-case perturbations of each training image within a ball defined by the attack model. In this work, we study the general question of how to train classifiers on augmented training data without causing an increase in test error, while simultaneously preserving the benefits of augmentation such as robustness.

First, we analyze why some forms of data augmentation such as with adversarial $\ell_\infty$ perturbations increase error. Previous works (Tsipras et al., 2019; Zhang et al., 2019; Fawzi et al., 2018; Nakkiran, 2019) provide simple constructions to explain the increase in test error with adversarial perturbations, but rely on assumptions, such as incorrect labeling of the adversarial perturbations or insufficient complexity of the hypothesis class, that we do not expect to hold in practice. We seek a simple theoretical setup that can shed light on why augmentation, even with label-preserving perturbations in a well-specified setting, causes an increase in test error. On the surface, it seems like we have only added information about the label distribution, so why does the test error increase?

In this work, we theoretically study minimum norm interpolation in well-specified linear regression, and show that data augmentation with label-preserving perturbations can increase the test error in some regimes, even when the targets are noiseless. For example, Figure 1(a) illustrates a function interpolation problem via cubic splines which exemplifies this phenomenon. Without data augmentation, the estimated predictor (dashed blue) is a line that captures the global structure and obtains low error. Data augmentation with local perturbations (crosses) encourages the predictor to fit the local structure of the high density points but compromises the global structure on the tail (solid orange) (Figure 1(b)). We show that this tension between local and global fit stems from the estimator having the wrong inductive bias. In particular, the minimum norm estimator minimizes a generic parameter norm while the test error is measured by a possibly different norm on the parameter error vector which depends on the data distribution (Section 4.1). Further, one might expect augmentation to be most helpful in low data settings. We show that in linear regression, this is also exactly the regime where augmentation can be most harmful. On real datasets, we similarly observe that data augmentation can be more detrimental with a smaller original training set (Section 5).

Motivated by our analysis of interpolation in linear regression, we propose a new estimator for data augmentation based on *X-regularization* (Section 6). X-regularization encourages the data augmented

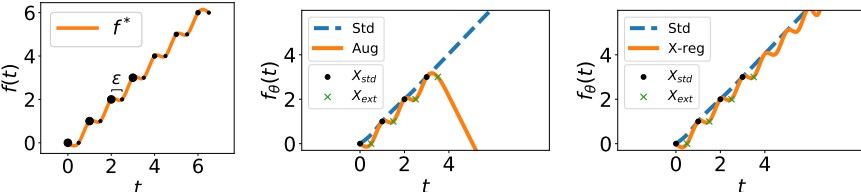

Figure 1: We consider perfectly fitting training points using cubic splines while minimizing *smoothness* of the function. **(Left)** depicts the true function $f^\star$ and the mass $P_{\mathsf{xy}}$ on each point $(x, y)$ size of the circles) **(Middle)** With a small number of standard training samples (circles) from the distribution depicted in (a), augmentating the dataset with local perturbations (crosses) causes the augmented estimator to have larger error than the standard estimator due to being maximally smooth while also fitting the augmented local perturbations. **(Right)** Our proposed X-regularization regularizes the model predictions towards the predictions of a standard model (trained without perturbations) on unlabeled data, allowing for fitting both the local and global structure.

interpolant to stay close to the original interpolant while fitting the extra augmented points. We prove that X-regularization eliminates the increase in test error upon data augmentation in the case of noiseless linear regression. See Figure 1(c) for its effect on the spline interpolation problem.

X-regularization naturally extends to more general losses and complex models and is closely related to self-training (Scudder, 1965), the classical semisupervised learning algorithm. For the particular setting of robustness, X-regularization applied to adversarial training takes the form of robust self-training (RST) that was recently proposed in (Carmon et al., 2019; Najafi et al., 2019; Uesato et al., 2019). The previous works view RST as a way to beat the sample complexity barrier of robustness. By showing that RST is an instantiation of X-regularization, we motivate RST as an appropriate data dependent regularizer that improves both standard accuracy and robustness. We evaluate the effect of RST on standard and robust accuracy with different adversarial training losses and perturbations on CIFAR-10 in Section 6.3. With $\ell_\infty$ perturbations, we find that RST improves standard accuracy by $4-6\%$ while maintaining or even improving the robustness achieved by the vanilla adversarial training counterpart. With random and adversarial rotations, RST improves standard accuracy by $\sim 1\%$ and robust accuracy by $1-3\%$. Our experiments suggest that RST, and more broadly using unlabeled data with X-regularization, is a promising approach to mitigate the undesirable drop in standard accuracy when training robust networks.

## 2 RELATED WORK

The detrimental effect of data augmentation has been previously studied in the context of a "tradeoff" between accuracy and robustness.

**Understanding the tradeoff.** In an attempt to explain the tradeoff between robustness and accuracy, Tsipras et al. (2019); Zhang et al. (2019); Fawzi et al. (2018); Nakkiran (2019) provide simple constructions that showcase an inherent tension between these objectives even in the limit of infinite data. These constructions rely on either non-label-preserving perturbations or insufficient model complexity to express a robust and accurate classifier. However, in practice, we typically augment with "imperceptible" perturbations that do not change the label, and we assume large neural networks used in practice to be expressive enough to contain a robust and accurate classifier. We address both these insufficiencies by studying covariate-shifted data augmentation where the extra data is labeled according to the same predictive distribution as the original data, and well-specified linear regression.

**Mitigating the tradeoff.** Existing proposals to mitigate the observed increase in standard error caused by adversarial augmentations are based on finding better architectures via Neural Architecture Search (Cubuk et al., 2017) or changing the neural net training algorithm (Lamb et al., 2019). While these methods have shown some success, they are restricted to neural networks. We complement this line of work by studying this tradeoff more generally and also provide some theoretical justification.

## 3 SETUP

### 3.1 COVARIATE-SHIFTED DATA AUGMENTATION.

Let $P_{xy}$ denote the underlying distribution of $(x,y)$ pairs, $P_x$ its marginal on $\mathbb{R}^d$, and $P_y(\cdot \mid x)$ the conditional distribution of the targets given inputs.

We refer to the training data from the underlying distribution $P_{xy}$ by *standard* training data. Formally, we have $n$ pairs $(x_i, y_i) \sim P_{xy}$ forming the measurement matrix $X_{std} = [x_1, x_2, ... x_n]^\top \in \mathbb{R}^{n \times d}$ and target vector $y_{std} = [y_1, y_2, ... y_n]^\top \in \mathbb{R}^n$.

Analogously, we consider augmenting the training set with "extra" training points denoted by $X_{ext} = [\tilde{x}_1, \tilde{x}_2, ... \tilde{x}_m]^\top \in \mathbb{R}^{m \times d}$ with associated targets $y_{ext} = [\tilde{y}_1, \tilde{y}_2, ... \tilde{y}_m]^\top \in \mathbb{R}^m$. We focus on covariate-shifted data augmentations which includes most forms of data augmentations used in practice. Here, the targets $y_{ext}$ are drawn from the same underlying predictive distribution $P_y(\cdot \mid x)$ as the standard data, while the extra inputs $X_{ext}$ could be arbitrary.

**Examples.** Typically, the "extra" training points are constructed as follows: $\tilde{x}_i = T(x_i)$, $\tilde{y}_i = y_i$, where $T$ is some label-preserving transformation thereby enforcing that the extra targets are as if sampled from $P_y(\cdot \mid x)$. Example transformations include translations, horizontal flips, small rotations, small $\ell_\infty$ perturbations in vision, and replacing words with their synonyms in NLP. However, our treatment of data augmentation in this work is more general where $X_{ext}$ is not necessarily obtained via transformations of $X_{std}$. Empirically, our main focus is on mitigating the increase in test error upon adversarial training. Most popular forms of adversarial training are instances of covariate-shifted data augmentation. Consider projected gradient adversarial training (PG-AT) (Madry et al., 2018) which confers robustness to adversarial examples (Szegedy et al., 2014) but also causes an increase in standard test error. PG-AT can be viewed as a form of iterative data augmentation that falls in our framework above. Formally, let $B(x)$ be the set of perturbations of $x$ that we would like to be robust against. We assume that the label is constant over $B(x)$. The transformation at step $t$ of the training for data point $x_i$ is $T_t(x_i) = \underset{x \in B(x_i)}{\operatorname{argmax}} \ell(\hat{\theta}_t, x_i, y_i)$, where $\hat{\theta}_t$ are the model parameters at time step $t$. The worst-case (max loss) perturbation is approximated via the projected gradient method.

### 3.2 MINIMUM NORM INTERPOLATION IN WELL-SPECIFIED LINEAR REGRESSION

Consider a regression task where the targets $y \in \mathbb{R}$ are drawn from the conditional distribution $P_y(\cdot \mid x) = \mathcal{N}(x^\top \theta^\star, \sigma)$, for some vector $\theta^\star \in \mathbb{R}^d$. Our goal is to learn a linear predictor $f_\theta(x) = x^\top \theta$.

In this work, we focus on interpolating estimators, which draws motivation from modern machine learning models that achieve near zero training loss (on both standard and extra augmented points). Interpolating estimators for linear models are analyzed in many recent works (Ma et al., 2018; Belkin et al., 2018; Hastie et al., 2019; Liang & Rakhlin, 2018; Bartlett et al., 2019). We present our results for interpolating linear regression estimators with minimum Euclidean norm, but our analysis directly applies to more general Mahalanobis norms via suitable rotations. See Appendix A. Given inputs $X \in \mathbb{R}^{n \times d}$ and corresponding targets $Y \in \mathbb{R}^n$ as training data, we define the following minimum norm interpolation estimator as

$$\hat{\theta} = \underset{\theta}{\operatorname{argmin}} \Big\{ \|\theta\|_2 : X\theta = Y \Big\}. \tag{1}$$

In particular, we compare the following estimators: (i) the standard estimator $\hat{\theta}_{std}$ which has $[X_{std}, y_{std}]$ as training data and (ii) the data augmented estimator $\hat{\theta}_{aug}$ with $X = [X_{std}; X_{ext}], Y = [y_{std}; y_{ext}]$ as training data:

$$\hat{\theta}_{std} = \underset{\theta}{\operatorname{argmin}} \Big\{ \|\theta\|_2 : X_{std}\theta = y_{std} \Big\} \text{ and } \hat{\theta}_{aug} = \underset{\theta}{\operatorname{argmin}} \Big\{ \|\theta\|_2 : X_{std}\theta = y_{std}, X_{ext}\theta = y_{ext} \Big\}. \tag{2}$$

## 4 BIAS AND VARIANCE OF MINIMUM NORM INTERPOLANTS

We evaluate the two estimators described in Equation 2 using the error on a random sample $x_{test}$ drawn from $P_x$. Let $\Sigma$ be the covariance of $P_x$. We focus on the error conditioned on $X_{std}, X_{ext}$, which

decomposes into a bias and variance term as follows

$$R(\hat{\theta}) = \mathbb{E}[(x_{\text{test}}^\top(\hat{\theta}-\theta^\star))^2] = \mathbb{E}[(\hat{\theta}-\theta^\star)^\top \Sigma(\hat{\theta}-\theta^\star)]$$
$$= \underbrace{(\mathbb{E}[\hat{\theta}]-\theta^\star)^\top \Sigma(\mathbb{E}[\hat{\theta}]-\theta^\star)}_{\text{Bias } B(\hat{\theta})} + \underbrace{\operatorname{tr}(\operatorname{Cov}(\hat{\theta})\Sigma)}_{\text{Variance } V(\hat{\theta})}, \tag{3}$$

where the expectation is taken over the randomness in $x_{\text{test}}$ and targets $y_{\text{std}}, y_{\text{ext}}$ conditioned on $X_{\text{std}}, X_{\text{ext}}$. In this section, we treat $X_{\text{std}}, X_{\text{ext}}$ as fixed quantities. However, since $X_{\text{std}}$ consists of samples from $P_x$, the number of samples dictates some structure on $X_{\text{std}}$. We touch upon this aspect in Section **??** where we study how the effect of augmentation varies with the number of samples in the original training set, both theoretically and empirically.

## 4.1 BIAS OF MINIMUM NORM INTERPOLANTS

For minimum norm interpolant with $X, Y$ as training data, the bias term $B(\hat{\theta})$ can be expressed as follows. On expectation (over targets $Y$), any interpolating estimator recovers $\theta^\star$ in the column space of $X$, but is unconstrained in $\operatorname{Null}(X^\top X)$. The minimum norm interpolant sets the component in $\operatorname{Null}(X^\top X)$ to zero. Formally,

$$\mathbb{E}[\hat{\theta}] = (X^\top X)^\dagger X^\top X \theta^\star \implies B(\hat{\theta}) = \theta^{\star\top} \Pi_X^\perp \Sigma \Pi_X^\perp \theta^\star,$$

where $\Pi_X^\perp = \left(I - (X^\top X)^\dagger(X^\top X)\right)$ is the projection matrix onto $\operatorname{Null}(X^\top X)$.

We now compare the bias of the standard and data augmented estimators (defined in Equation 2). It is convenient to define $\Sigma_{\text{std}} = X_{\text{std}}^\top X_{\text{std}}$ and $\Sigma_{\text{aug}} = X_{\text{std}}^\top X_{\text{std}} + X_{\text{ext}}^\top X_{\text{ext}}$. Let $\Pi_{\text{std}}^\perp = I - \Sigma_{\text{std}}^\dagger \Sigma_{\text{std}}$ and $\Pi_{\text{aug}}^\perp = I - \Sigma_{\text{aug}}^\dagger \Sigma_{\text{aug}}$ be the projection matrices onto $\operatorname{Null}(\Sigma_{\text{std}})$ and $\operatorname{Null}(\Sigma_{\text{aug}})$ respectively. Then,

$$B(\hat{\theta}_{\text{std}}) = \theta^{\star\top} \Pi_{\text{std}}^\perp \Sigma \Pi_{\text{std}}^\perp \theta^\star \quad \text{and} \quad B(\hat{\theta}_{\text{aug}}) = \theta^{\star\top} \Pi_{\text{aug}}^\perp \Sigma \Pi_{\text{aug}}^\perp \theta^\star. \tag{4}$$

**Remark 1.** *Note that the bias in parameter error is $\|\mathbb{E}[\hat{\theta}] - \theta^\star\|_2^2 = \|\Pi_X^\perp \theta^\star\|_2^2$. Since $Null(\Sigma_{aug}) \subseteq Null(\Sigma_{std})$, we have $\|\Pi_{aug}^\perp \theta^\star\|_2^2 \leq \|\Pi_{std}^\perp \theta^\star\|_2^2$, and data augmentation always reduces the bias in parameter error. However, the bias in test error upon augmentation $B(\hat{\theta}_{aug})$ could be larger or smaller than $B(\hat{\theta}_{std})$.*

### 4.1.1 SIMPLE LINEAR PROBLEM IN $\mathbb{R}^3$ WHERE ADDING DATA INCREASES BIAS

The following example illustrates how the interaction between the column spaces of standard inputs $X_{\text{std}}$, extra inputs $X_{\text{ext}}$ and the underlying true parameter $\theta^\star$ could cause data augmentation to increase bias.

For simplicity, we choose $\Sigma = \operatorname{diag}([\lambda_1, \lambda_2, \lambda_3])$ with $\lambda_2 \gg \lambda_1$, $X_{\text{std}} = e_3$ and additional data $X_{\text{ext}} = e_1 + e_2$ where $e_1, e_2, e_3$ denote the standard bases in $\mathbb{R}^3$. For brevity in notation, we denote $\mathbb{E}[\hat{\theta}_{\text{std}}]$ by $\hat{\theta}_{\text{std}}$ and $\mathbb{E}[\hat{\theta}_{\text{aug}}]$ by $\hat{\theta}_{\text{aug}}$.

Recall that by virtue of being minimum norm interpolants, $(\hat{\theta}_{\text{std}} - \theta^\star) \in \operatorname{Null}(\Sigma_{\text{std}}) = \mathbb{R}^2$ and $(\hat{\theta}_{\text{aug}} - \theta^\star) \in \operatorname{Null}(\Sigma_{\text{aug}}) = \{\rho(e_1 - e_2) : \rho \in \mathbb{R}\}$. Figure 2 depicts these parameter errors for different choices of $\theta^\star$.

**Bias under different settings of $\theta^\star$.** Plugging these terms into the bias expression in Equation (4) yields

$$B(\hat{\theta}_{\text{std}}) = \theta_1^{\star 2} \lambda_1 + \theta_2^{\star 2} \lambda_2 \quad \text{and} \quad B(\hat{\theta}_{\text{aug}}) = (1/4)(\theta_1^\star - \theta_2^\star)^2 \lambda_1 + (1/4)(\theta_1^\star - \theta_2^\star)^2 \lambda_2.$$

Since in our construction of $\Sigma$ we have $\lambda_2 \gg \lambda_1$, the bias expression is dominated by the coefficient on $\lambda_2$, which is the projection of the parameter error on $e_2$ (red lines in Figure 2). Depending on $\theta^\star$, $B(\hat{\theta}_{\text{aug}})$ could be larger or smaller than $B(\hat{\theta}_{\text{std}})$. In particular,

(i) when $\theta_1^\star \gg \theta_2^\star$ as in Fig. 2 (a), augmenting with $X_{\text{ext}}$ can increase bias $B(\hat{\theta}_{\text{aug}}) \gg B(\hat{\theta}_{\text{std}})$. Even though the augmented estimator has lower parameter error overall ($\|\hat{\theta}_{\text{aug}} - \theta^\star\|_2 \leq \|\hat{\theta}_{\text{std}} - \theta^\star\|_2$), the increase in parameter error along $e_2$ dominates the effect on the bias because $\lambda_2 \gg \lambda_1$.

(ii) when $\theta_2^\star \gg \theta_1^\star$ as in Fig. 2 (b), the same $X_{\text{ext}}$ causes $B(\hat{\theta}_{\text{aug}})$ to be smaller than $B(\hat{\theta}_{\text{std}})$. Here the augmented estimator has smaller parameter error along $e_2$ and hence decreasing bias despite an increase along $e_1$.

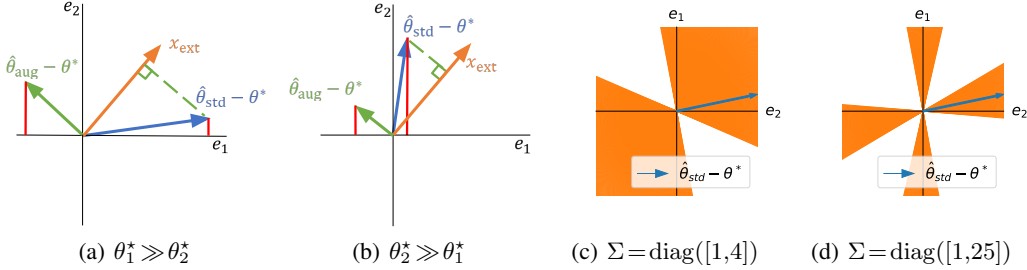

(a) $\theta_1^\star \gg \theta_2^\star$  (b) $\theta_2^\star \gg \theta_1^\star$  (c) $\Sigma = \text{diag}([1,4])$  (d) $\Sigma = \text{diag}([1,25])$

Figure 2: Illustration of the 3-D example described in Sec. 4.1.1. In **(a), (b)** we depict the errors $\hat{\theta}_{\text{aug}} - \theta^\star$ (green solid) and $\hat{\theta}_{\text{std}} - \theta^\star$ (blue solid) projected onto $\text{Null}(\Sigma_{\text{std}})$ that is spanned by eigenvectors $e_1$ and $e_2$ of $\Sigma$ with $\lambda_2 \gg \lambda_1$. Red lines depict the projection of parameter errors on $e_2$ which determines the bias. In (a) $\theta_1^\star \gg \theta_2^\star$ and bias increases upon augmentation. In (b) $\theta_1^\star \gg \theta_2^\star$ and bias decreases upon augmentation. **(c), (d)**The space of safe augmentation directions (orange) that don't increase bias for a given $\theta^\star$ are cone-shaped, where the cone width depends on the alignment of $\theta^\star$ with eigenvectors of $\Sigma$ and the skew in eigenvalues of $\Sigma$.

In summary, the minimum norm interpolant treats all unobserved dimensions in the null space "equally". In contrast, the bias is dominated by the components of the parameter error along the top eigenvectors of $\Sigma$. This mismatch could lead to settings where decreasing the null space of the training points via augmentation increases the error along top eigenvectors of $\Sigma$. We formalize this intuition and present a general characterization below.

### 4.1.2 GENERAL CHARACTERIZATIONS

We now study the biases of the standard and augmented estimators in general. Recall that $\Pi_{\text{std}}^\perp$ and $\Pi_{\text{aug}}^\perp$ are the projection matrices onto $\text{Null}(\Sigma_{\text{std}})$ and $\text{Null}(\Sigma_{\text{aug}})$ respectively, where $\Sigma_{\text{std}} = X_{\text{std}}^\top X_{\text{std}}$ and $\Sigma_{\text{aug}} = X_{\text{std}}^\top X_{\text{std}} + X_{\text{ext}}^\top X_{\text{ext}}$. Since $\text{Null}(\Sigma_{\text{aug}}) \subseteq \text{Null}(\Sigma_{\text{std}})$, we can decompose $\Pi_{\text{std}}^\perp \theta^\star$ into orthogonal components $v = \Pi_{\text{aug}}^\perp \theta^\star$ and $w = \Pi_{\text{std}}^\perp \Pi_{\text{aug}} \theta^\star$. Substituting this decomposition into Equation 4, we get the following exact characterization for when data augmentation increases bias.

**Theorem 1.** *The augmented estimator $\hat{\theta}_{aug}$ has larger bias i.e., $B(\hat{\theta}_{aug}) > B(\hat{\theta}_{std})$ if and only if*

$$v^\top \Sigma v < -2w^\top \Sigma v, \tag{5}$$

*where $v = \Pi_{std}^\perp \Pi_{aug} \theta^\star$ and $w = \Pi_{aug}^\perp \theta^\star$.*

The proof of Theorem 1 is in Appendix B.1. We see that condition (5) depends on $\theta^\star$ which is typically unknown. Hence, we cannot determine apriori if a particular form of augmentation would be "safe" and not increase error (like random translations in CIFAR-10) or harmful (like random rotations in CIFAR-10). However, we can make the following statements about when data augmentation is safe (for any $\theta^\star$) in some restricted settings.

1. When $\Sigma = I$, the condition (5) is always met (since $w \perp v$) and hence data augmentation is *always safe* and never increases bias for any $\theta^\star$. This suggests that data augmentation increases bias when there is a mismatch between the norm being minimized during interpolation and the norm of the parameter error that determines test error.

2. When $X_{\text{ext}}$ spans the entire nullspace of $\Sigma_{\text{std}}$ such that $\Sigma_{\text{aug}}$ is invertible, $w = 0$ for all $\theta^\star$ and data augmentation never increases bias.

3. In the simple case where $X_{\text{ext}}$ is rank-one, data augmentation is safe for all $\theta^\star$ if and only if $\Pi_{\text{std}}^\perp X_{\text{ext}}$ is an eigenvector of $\Sigma$. See Appendix B.4 for a proof.

Finally, we illustrate the safe augmentation directions $X_{\text{ext}}$ in the nullspace of $\Sigma_{\text{std}}$ for the simple 3-D problem discussed above for two different choices of $\Sigma$ and a fixed $\theta^\star$ (Figure 2 (c), (d)). The safe augmentations lie in cones around the eigenvectors of $\Sigma$ while the width and alignment of the cones depends on the alignment between $\theta^\star$ and the eigenvectors of $\Sigma$. As the eigenvalues of $\Sigma$ become more skewed,

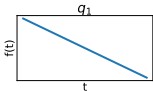 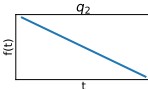 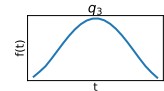 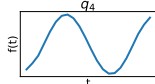

Figure 3: Top 4 eigenvectors of $\Sigma$ in the splines problem, representing wave functions in the input space. The eigenvalues corresponding to "global" eigenvectors which vary less over the domain are larger, making errors in global dimensions costly in terms of test error.

the space of safe augmentations shrinks. We present a dual perspective on Theorem 1 in Appendix B which characterizes the effect of augmentation on the bias in terms of properties of the true parameter $\theta^\star$.

**Local vs. global structure.** Finally, we tie our analysis back to the spline staircase problem from Figure 1. The inputs can be appropriately rotated so that the cubic spline interpolant is the minimum Euclidean norm interpolant (as in Equations 2). Under this rotation, the different eigenvectors of $\text{Null}(\Sigma_{\text{std}})$ measure either the fit in the "local" high frequency components or "global" low frequency components (See Figure 3). Any augmentation that encourages fitting local components in $\text{Null}(\Sigma_{\text{std}})$ could lead to a worse fit of the global structure, leading to increased test error (See Figure 3). This is suggestive of a similar trade-off phenomenon in practice where adversarial training (with say $\ell_\infty$ perturbations) encourages neural networks to fit the high-frequency components of the signal while compromising on the overall global structure causing an increase in test error.

## 4.2 Variance of Minimum Norm Interpolants

The main focus of this work is on the effect of data augmentation on the bias of minimum norm interpolants. For completeness, we present some conditions under which the data augmentation increases or decreases variance. For a more complete treatment, please refer to Appendix C. Let $X_{\text{std}}, y_{\text{std}}$ be the standard training data, with extra points $X_{\text{ext}}, y_{\text{ext}}$ and let $\Pi_{\text{std}}^\perp$ denote the projection matrix onto $\text{Null}(X_{\text{std}}^\top X_{\text{std}})$.

**Theorem 2.** *For the minimum norm interpolants defined in Equation 2, the following hold.*

1. *When $\Pi_{std}^\perp X_{ext} = 0$ such that the extra points lie in the column space of original standard training data, $V(\hat{\theta}_{aug}) \leq V(\hat{\theta}_{std})$.*

2. *When $X_{ext} \perp X_{std}$, such that the extra points lie entirely in $\text{Null}(X_{std}^\top X_{std})$, we have $V(\hat{\theta}_{aug}) \geq V(\hat{\theta}_{std})$.*

In general, when $X_{\text{ext}} \in \text{Null}(X_{\text{std}}^\top X_{\text{std}})$, we see that both the bias and variance of minimum norm interpolants could increase upon data augmentation, shedding some light on why data augmentation sometimes increases standard error in practice.

## 5 Effect of Size of the Original Training Set

In this section, we study how the effect of data augmentation varies as we vary the size of the original standard training set. We first briefly study this in the setting of minimum norm interpolation in linear regression. We then empirically evaluate the effect of adversarial training as we vary the number of original training samples in CIFAR-10 and observe that the empirical trends mirror the trends dictated by our analysis in linear regression.

### 5.1 Minimum norm interpolation—small and large data regimes.

Without loss of generality, we assume that $\Sigma$, the population covariance of $P_x$ is invertible since the test error only depends on columnspace of $\Sigma$.

**Large data regime.** In the large data regime where the number of standard training points $n \to \infty$, the empirical covariance of the original training points $\Sigma_{\text{std}} = X_{\text{std}}^\top X_{\text{std}} \approx n\Sigma$ is invertible with $\Pi_{\text{std}}^\perp = 0$. Both $\hat{\theta}_{\text{std}}$ and $\hat{\theta}_{\text{aug}}$ are unbiased (from Equation 4). From Theorem 2, variance of $\hat{\theta}_{\text{aug}}$ is never larger than

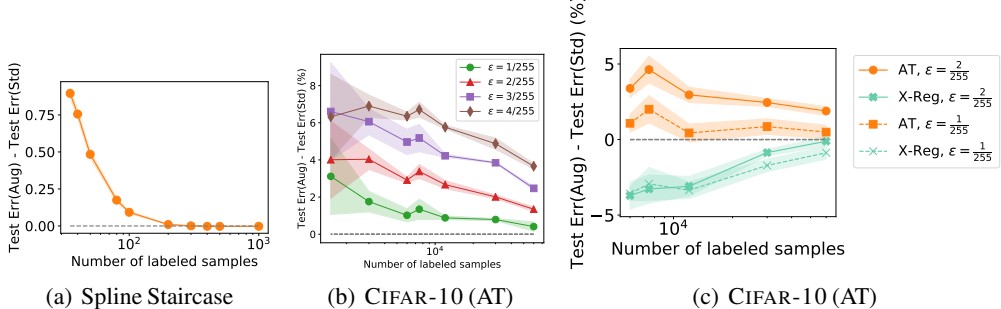

(a) Spline Staircase          (b) CIFAR-10 (AT)          (c) CIFAR-10 (AT)

Figure 4: Effect of data augmentation on test error as we vary the number of training samples. We plot the difference in errors of the augmented estimator and standard estimator. In both the spline staircase simulations and data augmentation with adversarial $\ell_\infty$ perturbations via adversarial training (AT) on CIFAR-10, we find that as we increase the number of samples, the increase in test error decreases.

that of $\hat{\theta}_{std}$. Putting together, the test error never increases upon augmentation in the large sample regime matching the common intuition that more data from the correct target distribution should never hurt.

**Small data regime.** In the small data regime, where $n$ is much smaller than $d$, the empirical covariance $\Sigma_{std}$ could be far from invertible. As we increase the number of samples, the null space $\text{Null}(\Sigma_{std})$ shrinks. Our analysis in Section 4 shows that for a fixed $\theta^\star$, the magnitude of possible increase in both bias and variance decreases as $\Pi_{std}^\perp X_{ext}$ decreases (See Appendix E) for details). This suggests that as the size of $X_{std}$ increases, the increase in test error due to augmentation decreases. We run simulations of the spline staircase example from Figure 1 and find that this trend holds (Figure 4(a); see Appendix D).

## 5.2 EMPIRICAL OBSERVATIONS ON THE EFFECT OF SAMPLE SIZE.

Do the trends in linear regression also hold for classification with more complex models and real world datasets? For our empirical study, we focus on adversarial training (Madry et al., 2018), iterative data augmentations with adversarial $\ell_\infty$ perturbations of different magnitudes $\epsilon$. We train a WideResNet-40-2 (Zagoruyko & Komodakis, 2016) on CIFAR-10 training set subsampled by varying amounts. We plot the difference in the test errors of the augmented and standard estimators as a function of training set size in Figure 4. We find that augmentation is less detrimental to test error with increase in sample size (Figure 4(b)), matching the trends predicted by our analysis of linear regression. Extrapolating the plot, we see that there should be no tradeoff between robustness and accuracy in the infinite data limit—contradicting the toy examples studied in (Tsipras et al., 2019; Zhang et al., 2019; Nakkiran, 2019) which we discussed in Section 2.

## 6 MITIGATING THE INCREASE IN BIAS UPON AUGMENTATION

To this point, the paper focuses on understanding why data augmentation could increase test error by analysing the setting of minimum norm interpolation in linear regression. However, data augmentation (e.g., via adversarial perturbations) often comes with desirable benefits such as robustness of estimators. In this section, we leverage our understanding from linear regression to design a new estimator for interpolating augmented data that mitigates the increase in test error while preserving the benefits. To this end, we introduce *X-regularization*, and prove that X-regularization eliminates the increase in bias upon augmentation for noiseless linear regression (Section 6.1. In Section 6.2, we show that this estimator naturally generalizes to arbitrary losses and complex models and is closely connected to the classical semi-supervised self-training algorithm (Scudder, 1965). We empirically evaluate the performance of this general X-regularization on adversarial training in Section 6.3. In a nutshell, X-regularization causes a smaller increase in standard error while maintaining or simultaneously improving the robustness of neural networks trained on CIFAR-10.

### 6.1 X-REGULARIZATION FOR LINEAR REGRESSION

Our development considers a stylized setting of interpolation in linear regression with noiseless observations from a linear model, $y = x^\top \theta^\star$, where the dimension is (much) larger than the number of observations. Let $\theta_{\text{int-std}}$ interpolate the initial data, satisfying $X_{\text{std}}\theta_{\text{int-std}} = y_{\text{std}}$. We use $\theta_{\text{int-std}}$ to construct a new data augmented estimator $\hat{\theta}_{\text{x-aug}}$ that interpolates both the standard data and augmented extra data while satisfying $R(\hat{\theta}_{\text{x-aug}}) \leq R(\theta_{\text{int-std}})$.

Given $\Sigma$ and an initial interpolant $\theta_{\text{int-std}}$, we propose the *X-regularized* data augmentation estimator

$$\hat{\theta}_{\text{x-aug}} \in \underset{\theta}{\operatorname{argmin}} \left\{ (\theta - \theta_{\text{int-std}})^\top \Sigma (\theta - \theta_{\text{int-std}}) : X_{\text{std}}\theta = y_{\text{std}},\ X_{\text{ext}}\theta = y_{\text{ext}} \right\}. \tag{6}$$

The X-regularized estimator $\hat{\theta}_{\text{x-aug}}$ optimizes for small error on the labeled data $(X_{\text{std}}, y_{\text{std}})$ and $(X_{\text{ext}}, y_{\text{ext}})$ while keeping the predictions of $\hat{\theta}_{\text{x-aug}}$ close to those of $\theta_{\text{int-std}}$ over unlabeled inputs drawn from $P_x$. To motivate our estimator, recall our discussion on the effect of data augmentation on the bias (test error in noiseless case) of minimum norm interpolants in Section 4.1. By fitting extra dimensions of $X_{\text{ext}}$, the data augmented estimator could have a larger parameter error than $\theta_{\text{int-std}}$ in important directions of $\Sigma$, and consequently higher test error (Figure 2). A natural strategy to mitigate this increase, then, is to fit $X_{\text{ext}}$ while staying *close* to $\theta_{\text{int-std}}$ weighted by $\Sigma$, which leads to the estimator defined in Equation 6. This intuition can be formalized to prove that the X-regularized interpolant $\hat{\theta}_{\text{x-aug}}$ never has higher test error than $\theta_{\text{int-std}}$.

**Theorem 3.** *Assume the noiseless linear model $y = x^\top \theta^\star$. Let $\theta_{\text{int-std}}$ be an arbitrary interpolant of the standard data, i.e. $X_{std}\theta_{\text{int-std}} = y_{std}$. Let $\hat{\theta}_{\text{x-aug}}$ be the X-regularized interpolant* (6). *Then*

$$R(\hat{\theta}_{\text{x-aug}}) \leq R(\theta_{\text{int-std}}).$$

See Appendix **??** for proof. To provide some graphical intuition for the result, consider the spline interpolant $\hat{\theta}_{\text{std}}$ Fig. 1 illustrates where $X_{\text{ext}}$ consists of local perturbations. The X-regularized estimator matches the standard interpolant $\hat{\theta}_{\text{std}}$ on points outside the training set, thereby capturing global structure while simultaneously fitting local structure on the training set via $X_{\text{ext}}$.

Note that our discussion on the effect of sample sizes in Section 5 suggests that a larger labeled standard training set would mitigate the drop in standard error due to augmentation. However, our development of X-regularization suggests that we only need *unlabeled* data, which is much cheaper to obtain in practice. Unlabeled data is often used to improve standard test error in the semi-supervised learning paradigm. Here, we motivate the use of unlabeled data to mitigate the possible increase in test error from data augmentation.

### 6.2 ROBUST SELF-TRAINING AS X-REGULARIZATION FOR ROBUSTNESS

To motivate using X-regularization for general models and losses, note that the expression of the objective function for the X-regularized estimator in (6) can be rewritten in a more general form:

$$(\theta - \theta_{\text{int-std}})^\top \Sigma (\theta - \theta_{\text{int-std}}) = \mathbb{E}_{P_x}[(x^\top \theta - x^\top \theta_{\text{int-std}})^2] = \mathbb{E}_{P_x}[\ell_{\text{sq}}(f_\theta(x), f_{\theta_{\text{int-std}}}(x))],$$

where $\ell_{\text{sq}}$ is the squared loss between the predictions of the model $f_\theta(x)$ with the predictions (pseudo-labels) of a given interpolant $f_{\theta_{\text{int-std}}}(x)$. A generalized version of X-regularization replaces $\ell_{\text{sq}}$ with some general loss $\ell$ that include classification losses such as the logistic loss. Written this way, X-regularization regularizes the predictions of an augmented estimator towards the predictions of the standard interpolant, similarly to the classical semi-supervised self-training algorithm (Scudder, 1965).

The main motivation of our work is to fix the the drop in standard test performance from augmentation strategies that seek to enforce robustness by adding label-preserving transformations of existing inputs. With augmentations take the form of some label-preserving transformations $T$, it is natural to consider transformations of both the labeled and unlabeled data as the set of "extra" points, that constitute $X_{\text{ext}}$. Generalizing the linear regression estimator of Equation 6 to use arbitrary losses and transformation

| Method | Robust Test Acc. | Standard Test Acc. |
|---|---|---|
| Standard Training | 0.8% | 95.2% |
| Standard Self-Training | 0.3% | 96.4% |
| PG-AT (Madry et al., 2018) | 45.8% | 87.3% |
| RST + PG-AT | **58.5%** | **91.8%** |
| TRADES (Zhang et al., 2019) | 55.4% | 84.0% |
| RST + TRADES (Carmon et al., 2019) | **63.1%** | **89.7%** |
| Interpolated AT (Lamb et al., 2019)[1] | 45.1% | 93.6% |
| Neural Arch. Search (Cubuk et al., 2017) | 50.1% | 93.2% |

| Method | Robust Test Acc. | Standard Test Acc. |
|---|---|---|
| Standard Training | 0.2% | 94.6% |
| Worst-of-10 | 73.9% | 95.0% |
| RST + Worst-of-10 | **75.1%** | **95.8%** |
| Random | 67.7% | 95.1% |
| RST + Random | **70.9%** | **95.8%** |
| Worst-of-10 (Engstrom et al., 2019)[2] | 69.2% | 91.3% |
| Random (Yang et al., 2019)[3] | 58.3% | 91.8% |

Table 1: Performance of X-regularization instantiated as robust self-training (RST) applied to different perturbations and adversarial training algorithms. **(Left)** CIFAR-10 standard test accuracy and robust test accuracy against $\ell_\infty$ perturbations of size $\epsilon = 8/255$. All methods use $\epsilon = 8/255$ while training. Robust accuracies are against a PG based attack with 20 steps. **(Right)** CIFAR-10 standard test accuracy and robust test accuracy against a grid attack of rotations up to 30 degrees and translations up to $\sim 10\%$ of the image size, following (Engstrom et al., 2019). All adversarial and random methods use rotations and translations with these same parameters during training. For both tables, all shaded rows make use of 500K unlabeled images from 80M Tiny Images sourced in (Carmon et al., 2019). RST improves *both* the standard and robust accuracy over the vanilla counterparts for different algorithms (AT and TRADES) as well as different perturbations ($\ell_\infty$ and rotation/translations).

based augmentations, we get,

$$\hat{\theta}_{\text{x-aug}} := \underset{\theta}{\arg\min} \Bigg\{ N^{-1} \sum_{(x,y) \in [X_{\text{std}}, y_{\text{std}}]} \ell(f_\theta(x), y) + \beta \, \tilde{\ell}(f_\theta(T(x)), y)$$
$$+ \lambda m^{-1} \sum_{i=1}^{m} \ell(f_\theta(\tilde{x}_i), f_{\hat{\theta}_{\text{std}}}(\tilde{x}_i)) + \beta \, \tilde{\ell}(f_\theta(T(\tilde{x}_i)), f_{\hat{\theta}_{\text{std}}}(\tilde{x}_i)) \Bigg\}. \qquad (7)$$

We note that the general X-regularized estimator above is the Robust Self-Training (RST) algorithm proposed and studied recently by Carmon et al. (2019), when applied to arbitrary transformations. Variants of RST were also studied in (Najafi et al., 2019; Uesato et al., 2019). By deriving RST via X-regularization, we provide some theoretical justification for why RST would improve standard accuracy.

### 6.3 EMPIRICAL EVALUATION OF X-REGULARIZATION AS RST

For our empirical investigation, we evaluate the effect of X-regularization on the commonly observed tradeoff between robustness and standard accuracy when augmenting the training set with transformations. We recall that X-regularization when applied to transformation based augmentations leads to Robust Self Training. Therefore, we refer to X-regularization and RST synonymously through this section. We instantiate the robust self-training estimator defined in Equation 7 on both PG-AT and TRADES; exact loss functions appear in Appendix F.2. All of our experiments are on CIFAR-10, and RST estimators use 500K unlabeled images sourced from Tiny Images in (Carmon et al., 2019) in addition to the labeled training set.

In our first experiment, we use the same settings as our experiments on effect of sample size in CIFAR-10 in Section 5. We compare the error of RST+PG-AT and standard training in Figure 4. We find that RST+PG-AT has *lower* standard test error than the standard training, while simultaneously fitting the training data robustly and achieving higher robustness (see Appendix G.2.1). We also see the maximum gains from RST+PG-AT are in the small data regime where adversarial training has the most affect on standard error.

In our next experiment, we compare with other methods in the literature. We train a larger WRN-28-10 model with the entire labeled CIFAR-10 training set and 500K unlabeled images. Table 1(left) summarizes the results. While there is a still a drop in accuracy compared to standard training, RST has higher standard accuracy than the vanilla counterparts without sacrificing robustness, in both PG-AT and TRADES. The gains are comparable to gains from other measures to improve the tradeoff between

robustness and accuracy such as Interpolated Adversarial Training and Neural Architecture Search. RST could be integrated with the above training algorithms to see further gains; we leave this to future work.

Finally, we test the effect of RST on a different family of perturbations. We consider adversarial (worst of 10) and random augmentations using simultaneous rotations and translations of the input image. Table 1(right) presents the results. While the augmented estimators marginally improve standard accuracy for these perturbations, applying RST increases both robust and standard accuracies beyond that of the augmented estimator in both cases. This shows that RST is beneficial even when data augmentation does not decrease standard test error.

## 7 DISCUSSION

**Semi-supervised setting.** General X-regularization leverages unlabeled data to mitigate the possible harmful effect of data augmentation. The traditional setup of semi-supervised learning involves only one objective: improve the standard accuracy on the underlying population. We revisit the semi-supervised setting with a different focus. For several applications, standard supervised deep learning provides highly accurate classifiers, but they are surprisingly brittle. Attempts to improve robustness typically lower accuracy. Training robust *and* accurate classifiers remains an open challenge, and semi-supervised learning is emerging as a promising approach. Recent work (Carmon et al., 2019; Najafi et al., 2019; Uesato et al., 2019) has studied the benefits of semi-supervised learning in improving robustness. In our work, we bolster this line of work by demonstrating that semi-supervised learning can simultaneously improve the accuracy, while maintaining robustness.

**Self-training.** In this work, we study X-regularization which is closely related to self-training, perhaps the oldest semi-supervised learning algorithm (Scudder, 1965). In the traditional setting of improving standard accuracy, self-training has shown some success, but other approaches perform significantly better; see survey (Oliver et al., 2018). However, in the regime where we care about both robustness and accuracy, we see that self-training based approaches such as X-regularization offer significant benefits. We provide a detailed comparison of X-regularization to other semi-supervised learning algorithms in Appendix H. Variants of self-training are also gaining prominence in the related but different setting of unsupervised domain adaptation. Here, the unlabeled data is from the "target" distribution, while the labeled data is from a different "source" distribution and the goal is to perform well on the target distribution.

**Interpolation in linear regression.** Analysis of the interpolation regime has recently gained prominence with the observation that neural networks obtain zero training error. Previous works (Hastie et al., 2019; Bartlett et al., 2019; Belkin et al., 2019) study the performance of minimum norm interpolants in overparameterized linear regression in an attempt to explain the generalization properties of neural networks that are not explained by the classical perspective on interpolation and overfitting. In our work, we show that the same overparameterized setting also sheds light on the empirical observation that data augmentation sometimes helps and sometimes harms test performance. In contrast, in the classical underparameterized regime, data augmentation never harms test performance as common statistical intuition would suggest. Studying interpolation in the overparameterized regime even in simple settings such as linear regression thus seems to be valuable in understanding the properties of neural networks in practice.

**Conclusion.** We studied adversarial training through the lens of data augmentation with the goal of training robust and accurate classifiers. We analyzed general data augmentation in a stylized setting and proved that unlabeled data can eliminate possible increase in test error. This motivated a general estimator based on self-training combined with adversarial training that shows promise in improving both the accuracy and robustness of neural networks in practice. While using unlabeled data via simple self-training has shown to improve both accuracy and robustness, how to best utilize unlabeled data in this context is an open question. Further, can we obtain highly robust and accurate networks by simply using a large amount of unlabeled data, or do we need further innovations in neural network architectures and training?

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

## A  TRANSFORMATIONS TO HANDLE ARBITRARY MATRIX NORMS

Consider a more general minimum norm estimator of the following form. Given inputs $X$ and corresponding targets $Y$ as training data, we study the interpolation estimator,

$$\hat{\theta} = \operatorname*{argmin}_{\theta}\left\{\theta^{\top}M\theta : X\theta = Y\right\}, \tag{8}$$

where $M$ is a positive definite (PD) matrix that incorporates prior knowledge about the true model. For simplicitly, we present our results in terms of the $\ell_2$ norm (ridgeless regression) as defined in Equation 8. However, all our results hold for arbitrary $M$–norms via appropriate rotations. Given an arbitrary PD matrix $M$, the rotated covariates $x \leftarrow M^{-1/2}x$ and rotated parameters $\theta \leftarrow M^{1/2}\theta$ maintain $Y = X\theta + \sigma\mathcal{N}(0,I)$ and the $M$-norm of parameters simplifies to $\|\theta\|_2$.

## B  BIAS OF MINIMUM NORM INTERPOLANTS

### B.1  PROOF OF THEOREM 1

Inequality (5) follows from

$$\begin{aligned}
B(\hat{\theta}_{\mathrm{aug}}) - B(\hat{\theta}_{\mathrm{std}}) &= (\theta^{\star} - \hat{\theta}_{\mathrm{aug}})^{\top}\Sigma(\theta^{\star} - \hat{\theta}_{\mathrm{aug}}) - (\theta^{\star} - \hat{\theta}_{\mathrm{std}})^{\top}\Sigma(\theta^{\star} - \hat{\theta}_{\mathrm{std}}) \\
&= (\Pi_{\mathrm{aug}}^{\perp}\theta^{\star})^{\top}\Sigma\Pi_{\mathrm{aug}}^{\perp}\theta^{\star} - (\Pi_{\mathrm{std}}^{\perp}\theta^{\star})^{\top}\Sigma\Pi_{\mathrm{std}}^{\perp}\theta^{\star} \\
&= w^{\top}\Sigma w - (w+v)^{\top}\Sigma(w+v) \\
&= -2w^{\top}\Sigma v - v^{\top}\Sigma v
\end{aligned} \tag{9}$$

by decomposition of $\Pi_{\mathrm{std}}^{\perp}\theta^{\star} = v + w$ where $v = \Pi_{\mathrm{std}}^{\perp}\Pi_{\mathrm{aug}}\theta^{\star}$ and $w = \Pi_{\mathrm{std}}^{\perp}\Pi_{\mathrm{aug}}^{\perp}\theta^{\star}$. We also want to note that the bias difference does scale with $\|\theta^{\star}\|^2$.

## B.2 BIAS INCREASE REQUIRES COMPLEX TRUE ESTIMATORS

A dual perspective on Theorem 1 leads the following proposition that characterizes the properties of the true function $\theta^\star$ that leads to harmful augmentations.

**Proposition 1.** *For a given $X_{std}, X_{ext}, \Sigma$, a bias increase of $B(\hat{\theta}_{aug}) - B(\hat{\theta}_{std}) = c > 0$ via augmentation with $X_{ext}$ is possible only if $\theta^\star$ is sufficiently more complex than $\hat{\theta}_{std}$ in the $\ell_2$ norm, i.e.*

$$\|\theta^\star\|_2^2 - \|\hat{\theta}_{std}\|_2^2 > \gamma c \tag{10}$$

*for some scalar $\gamma > 0$ that depends on $X_{std}, X_{ext}, \Sigma$.*

### B.2.1 PROOF OF INEQUALITY (10)

The proof of inequality (10) is based on the following two lemmas that are also useful for characterization purposes in Corollary 1.

**Lemma 1.** *If a PSD matrix $\Sigma$ has non-equal eigenvalues, one can find two unit vectors $w, v$ for which the following holds*

$$w^\top v = 0 \qquad and \qquad w^\top \Sigma v \neq 0 \tag{11}$$

*Hence, there exists a combination of original and augmentation dataset $X_{std}, X_{ext}$ such that condition (11) holds for two directions $v \in Col(\Pi_{std}^\perp \Pi_{aug})$ and $w \in Col(\Pi_{std}^\perp \Pi_{aug}^\perp) = Col(\Pi_{aug}^\perp)$.*

Note that neither $w$ nor $v$ can be eigenvectors of $\Sigma$ in order for both conditions in equation (11) to hold. Given a population covariance, fixed original and augmentation data for which condition (11) holds, we can now explicitly construct $\theta^\star$ for which augmentation hurts bias.

**Lemma 2.** *Assume $\Sigma$, $X_{std}$, $X_{ext}$ are fixed. Then condition (11) holds for two directions $v \in Col(\Pi_{std}^\perp \Pi_{aug})$ and $w \in Col(\Pi_{std}^\perp \Pi_{aug}^\perp)$ iff there exists a $\theta^\star$ such that $B(\hat{\theta}_{aug}) - B(\hat{\theta}_{std}) \geq c$ for some $c > 0$. Furthermore, the $\ell_2$ norm of $\theta^\star$ needs to satisfy the following lower bounds with $c_1 := \|\hat{\theta}_{aug}\|^2 - \|\hat{\theta}_{std}\|^2$*

$$\|\theta^\star\|^2 - \|\hat{\theta}_{aug}\|^2 \geq \beta_1 c_1 + \beta_2 \frac{c^2}{c_1}$$

$$\|\theta^\star\|^2 - \|\hat{\theta}_{std}\|^2 \geq (\beta_1 + 1)c_1 + \beta_2 \frac{c^2}{c_1} \tag{12}$$

*where $\beta_i$ are constants that depend on $X_{std}, X_{ext}, \Sigma$.*

Inequality (10) follows directly from the second statement of Lemma 2 by minimizing the bound (12) with respect to $c_1$ which is a free parameter to be chosen during construction of $\theta^\star$ (see proof of Lemma (2). The minimum is attained for $c_1 = 2\sqrt{(\beta_1 + 1)(\beta_2 c^2)}$. We hence conclude that $\theta^\star$ needs to be sufficiently more complex than a good standard solution, i.e. $\|\theta^\star\|_2^2 - \|\hat{\theta}_{std}\|_2^2 > \gamma c$ where $\gamma > 0$ is a constant that depends on the $X_{std}, X_{ext}$.

## B.3 PROOF OF TECHNICAL LEMMAS

In this section we prove the technical lemmas that are used to prove Theorem 1.

### B.3.1 PROOF OF LEMMA 2

Any vector $\Pi_{std}^\perp \theta \in \text{Null}(\Sigma_{std})$ can be decomposed into orthogonal components $\Pi_{std}^\perp \theta = \Pi_{std}^\perp \Pi_{aug}^\perp \theta + \Pi_{std}^\perp \Pi_{aug} \theta$. Using the minimum-norm property, we can then always decompose the (rotated) augmented estimator $\hat{\theta}_{aug} \in Col(\Pi_{aug}^\perp) = Col(\Pi_{std}^\perp \Pi_{aug}^\perp)$ and true parameter $\theta^\star$ by

$$\hat{\theta}_{aug} = \hat{\theta}_{std} + \sum_{v_i \in \text{ext}} \zeta_i v_i$$

$$\theta^\star = \hat{\theta}_{aug} + \sum_{w_j \in \text{rest}} \xi_j w_j,$$

where we define "ext" as the set of basis vectors which span $\mathrm{Col}(\Pi_{\mathrm{std}}^{\perp}\Pi_{\mathrm{aug}})$ and respectively "rest" for $\mathrm{Null}(\Sigma_{\mathrm{aug}})$. Requiring the bias increase to be some constant $c > 0$ can be rewritten using identity (9) as follows

$$B(\hat{\theta}_{\mathrm{aug}}) - B(\hat{\theta}_{\mathrm{std}}) = c$$

$$\iff \left(\sum_{v_i \in \mathrm{ext}} \zeta_i v_i\right)^{\top} \Sigma \left(\sum_{v_i \in \mathrm{ext}} \zeta_i v_i\right) + c = -2\left(\sum_{w_j \in \mathrm{rest}} \xi_j w_j\right)\Sigma\left(\sum_{v_i \in \mathrm{ext}} \zeta_i v_i\right)$$

$$\iff \left(\sum_{v_i \in \mathrm{ext}} \zeta_i v_i\right)^{\top} \Sigma \left(\sum_{v_i \in \mathrm{ext}} \zeta_i v_i\right) + c = -2 \sum_{w_j \in \mathrm{rest}, v_i \in \mathrm{ext}} \xi_j \zeta_i w_j^{\top} \Sigma v_i \qquad (13)$$

The left hand side of equation (13) is always positive, hence it is necessary for this equality to hold with any $c > 0$, that there exists at least one pair $i, j$ such that $w_j^{\top} \Sigma v_i \neq 0$ and one direction of the iff statement is proved.

For the other direction, we show that if there exist $v \in \mathrm{Col}(\Pi_{\mathrm{std}}^{\perp}\Pi_{\mathrm{aug}})$ and $w \in \mathrm{Col}(\Pi_{\mathrm{std}}^{\perp}\Pi_{\mathrm{aug}}^{\perp})$ for which condition (11) holds (wlog we assume that the $w^{\top}\Sigma v < 0$) we can construct a $\theta^{\star}$ for which the inequality (5) in Theorem 1 holds as follows:

It is then necessary by our assumption that $\xi_j \zeta_i w_j^{\top} \Sigma v_i > 0$ for at least some $i, j$. We can then set $\zeta_i > 0$ such that $\|\hat{\theta}_{\mathrm{aug}} - \hat{\theta}_{\mathrm{std}}\|^2 = \|\zeta\|^2 = c_1 > 0$, i.e. that the augmented estimator is not equal to the standard estimator (else obviously there can be no difference in bias and equality (13) cannot be satisfied for any desired bias increase $c > 0$).

The choice of $\xi$ minimizing $\|\theta^{\star} - \hat{\theta}_{\mathrm{aug}}\|^2 = \sum_j \xi_j^2$ that also satisfies equation (13) is an appropriately scaled vector in the direction of $x = W^{\top}\Sigma V \zeta$ where we define $W := [w_1, \ldots, w_{|\mathrm{rest}|}]$ and $V := [v_1, \ldots, v_{|\mathrm{ext}|}]$. Defining $c_0 = \zeta^{\top} V^{\top}\Sigma V \zeta$ for convenience and then setting

$$\xi = -\frac{c_0 + c}{2\|x\|_2^2} x \qquad (14)$$

which is well-defined since $x \neq 0$, yields a $\theta^{\star}$ such that augmentation hurts. It is thus necessary for $B(\hat{\theta}_{\mathrm{aug}}) - B(\hat{\theta}_{\mathrm{std}}) = c$ that

$$\sum_j \xi_j^2 = \frac{(c_0 + c)^2}{4\|W^{\top}\Sigma V \zeta\|^2} = \frac{(\zeta^{\top} V^{\top}\Sigma V \zeta + c)^2}{4\zeta^{\top} V^{\top}\Sigma W W^{\top}\Sigma V \zeta}$$

$$\geq \frac{(\zeta^{\top} V^{\top}\Sigma V \zeta)^2}{4\zeta^{\top} V^{\top}\Sigma W W^{\top}\Sigma V \zeta} + \frac{c^2}{4\zeta^{\top} V^{\top}\Sigma W W^{\top}\Sigma V \zeta}$$

$$\geq \frac{c_1}{4}\frac{\lambda_{\min}^2(V^{\top}\Sigma V)}{\lambda_{\max}^2(W^{\top}\Sigma V)} + \frac{c^2}{4c_1\lambda_{\max}^2(W^{\top}\Sigma V)}.$$

By assuming existence of $i, j$ such that $\xi_j \zeta_i w_j^{\top} \Sigma v_i \neq 0$, we are guaranteed that $\lambda_{\max}^2(W^{\top}\Sigma V) > 0$.

Note due to construction we have $\|\theta^{\star}\|_2^2 = \|\hat{\theta}_{\mathrm{std}}\|_2^2 + \sum_i \zeta_i^2 + \sum_j \xi_j^2$ and plugging in the choice of $\xi_j$ in equation (14) we have

$$\|\theta^{\star}\|_2^2 - \|\hat{\theta}_{\mathrm{std}}\|_2^2 \geq c_1 \left[1 + \frac{\lambda_{\min}^2(V^{\top}\Sigma V)}{4\lambda_{\max}^2(W^{\top}\Sigma V)}\right] + \frac{c^2}{4\lambda_{\max}^2(W^{\top}\Sigma V)}\frac{1}{c_1}.$$

Setting $\beta_1 = \left[1 + \frac{\lambda_{\min}^2(V^{\top}\Sigma V)}{4\lambda_{\max}^2(W^{\top}\Sigma V)}\right]$, $\beta_2 = \frac{1}{4\lambda_{\max}^2(W^{\top}\Sigma V)}$ yields the result.

### B.3.2 PROOF OF LEMMA 1

Let $\lambda_1, \ldots, \lambda_m$ be the $m$ non-zero eigenvalues of $\Sigma$ and $u_i$ be the corresponding eigenvectors. Then choose $v$ to be any combination of the eigenvectors $v = U\beta$ where $U = [u_1, \ldots, u_m]$ where at least $\beta_i, \beta_j \neq$

0 for $\lambda_i \neq \lambda_j$. We next construct $w = U\alpha$ by choosing $\alpha$ as follows such that the inequality in (11) holds:

$$\alpha_i = \frac{\beta_j}{\beta_i^2 + \beta_j^2}$$

$$\alpha_j = \frac{-\beta_i}{\beta_i^2 + \beta_j^2}$$

and $\alpha_k = 0$ for $k \neq i,j$. Then we have that $\alpha^\top \beta = 0$ and hence $w^\top v = 0$. Simultaneously

$$w^\top \Sigma v = \lambda_i \beta_i \alpha_i + \lambda_j \beta_j \alpha_j$$

$$= (\lambda_i - \lambda_j) \frac{\beta_i \beta_j}{\beta_i^2 + \beta_j^2} \neq 0$$

which concludes the proof of the first statement.

We now prove the second statement by constructing $\Sigma_{\text{std}} = X_{\text{std}}^\top X_{\text{std}}, \Sigma_{\text{ext}} = X_{\text{ext}}^\top X_{\text{ext}}$ using $w, v$. We can then obtain $X_{\text{std}}, X_{\text{ext}}$ using any standard decomposition method to obtain $X_{\text{std}}, X_{\text{ext}}$. We construct $\Sigma_{\text{std}}, \Sigma_{\text{ext}}$ using $w, v$. Without loss of generality, we can make them simultaneously diagonalizable. We construct a set of eigenvectors that is the same for both matrices paired with different eigenvalues. Let the shared eigenvectors include $w, v$. Then if we set the corresponding eigenvalues $\lambda_w(\Sigma_{\text{ext}}) = 0, \lambda_v(\Sigma_{\text{ext}}) > 0$ and $\lambda_w(\Sigma_{\text{std}}) = 0, \lambda_v(\Sigma_{\text{std}}) = 0$, then $\lambda_w(\Sigma_{\text{aug}}) = 0$ such that $w \in \text{Col}(\Pi_{\text{std}}^\perp \Pi_{\text{aug}}^\perp)$ and $v \in \text{Col}(\Pi_{\text{std}}^\perp \Pi_{\text{aug}})$. This shows the second statement. With this, we can design a $\theta^\star$ for which augmentation hurts as in Lemma 2.

### B.4 CHARACTERIZATION COROLLARY 1

A simpler case to analyze is when we only augment with one extra data point. The following corollary characterizes which single augmentation directions lead to higher prediction error for the augmented estimator.

**Corollary 1.** *The following characterizations hold for augmentation directions that do not cause the bias of the augmented estimator to be higher than the original estimator.*

(a) (in terms of ratios of inner products) *For a given $\theta^\star$, data augmentation does not increase the bias of the augmented estimator for a single augmentation direction $x_{ext}$ if*

$$\frac{x_{ext}^\top \Pi_{std}^\perp \Sigma \Pi_{std}^\perp x_{ext}}{x_{ext}^\top \Pi_{std}^\perp x_{ext}} - 2\frac{(\Pi_{std}^\perp x_{ext})^\top \Sigma \Pi_{std}^\perp \theta^\star}{x_{ext}^\top \Pi_{std}^\perp \theta^\star} \leq 0 \tag{15}$$

(b) (in terms of eigenvectors) *Data augmentation does not increase bias for any $\theta^\star$ if $\Pi_{std}^\perp x_{ext}$ is an eigenvector of $\Sigma$. However if one augments in the direction of a mixture of eigenvectors of $\Sigma$ with different eigenvalues, there exists $\theta^\star$ such that augmentation hurts.*

(c) (depending on well-conditioning of $\Sigma$) *If $\frac{\lambda_{\max}(\Sigma)}{\lambda_{\min}(\Sigma)} \leq 2$ and $\Pi_{std}^\perp \theta^\star$ is an eigenvector of $\Sigma$, then no augmentations $x_{ext}$ increase bias.*

The form in Equation (15) compares ratios of inner products of $\Pi_{\text{std}}^\perp x_{\text{ext}}$ and $\Pi_{\text{std}}^\perp \theta^\star$ in two spaces: the one in the numerator is weighted by $\Sigma$ whereas the denominator is the standard inner product. Thus, if $\Sigma$ scales and rotates rather inhomogeneously, then augmenting with $x_{\text{ext}}$ may hurt bias. Here again, if $\Sigma = \gamma I$ for $\gamma > 0$, then the condition must hold.

### B.4.1 PROOF OF COROLLARY 1 (A)

Note that for a single augmentation point $X_{\text{ext}} = x_{\text{ext}}^\top$, the orthogonal decomposition of $\Pi_{\text{std}}^\perp \theta^\star$ into $\text{Col}(\Pi_{\text{aug}}^\perp)$ and $\text{Col}(\Pi_{\text{std}}^\perp \Pi_{\text{aug}})$ is defined by $v = \frac{\Pi_{\text{std}}^\perp x_{\text{ext}}^\top \theta^\star}{\|\Pi_{\text{std}}^\perp x_{\text{ext}}\|^2} \Pi_{\text{std}}^\perp x_{\text{ext}}$ and $w = \Pi_{\text{std}}^\perp \theta^\star - v$ respectively. Plugging back into into identity (9) then yields the following condition for safe augmentations:

$$2(v - \Pi_{\text{std}}^\perp \theta^\star)^\top \Sigma v - v^\top \Sigma v \leq 0 \tag{16}$$

$$v^\top \Sigma v - 2(\Pi_{\text{std}}^\perp \theta^\star)^\top \Sigma v \leq 0$$

$$\iff \Pi_{\text{std}}^\perp x_{\text{ext}}^\top \Sigma \Pi_{\text{std}}^\perp x_{\text{ext}} \leq 2(\Pi_{\text{std}}^\perp \theta^\star)^\top \Sigma \Pi_{\text{std}}^\perp x_{\text{ext}} \cdot \frac{\|\Pi_{\text{std}}^\perp x_{\text{ext}}\|^2}{\Pi_{\text{std}}^\perp x_{\text{ext}}^\top \theta^\star}$$

Rearranging the terms yields inequality (15).

Safe augmentation directions for specific choices of $\theta^\star$ and $\Sigma$ are illustrated in Figure 2.

### B.4.2 PROOF OF COROLLARY 1 (B)

Assume that $\Pi_{\text{std}}^\perp x_{\text{ext}}$ is an eigenvector of $\Sigma$ with eigenvalue $\lambda > 0$. We have

$$\frac{x_{\text{ext}}^\top \Pi_{\text{std}}^\perp \Sigma \Pi_{\text{std}}^\perp x_{\text{ext}}}{x_{\text{ext}}^\top \Pi_{\text{std}}^\perp x_{\text{ext}}} - 2\frac{(\Pi_{\text{std}}^\perp x_{\text{ext}})^\top \Sigma \Pi_{\text{std}}^\perp \theta^\star}{x_{\text{ext}}^\top \Pi_{\text{std}}^\perp \theta^\star} = -\lambda < 0$$

for any $\theta^\star$. Hence by Corollary 1 (a), the bias doesn't increase by augmenting with eigenvectors of $\Sigma$ for any $\theta^\star$.

When the single augmentation direction $v$ is not an eigenvector of $\Sigma$, by Lemma 1 one can find $w$ such that $w^\top \Sigma v \neq 0$. The proof in Lemma 1 gives an explicit construction for $w$ such that condition (11) holds and the result then follows directly by Lemma 2.

### B.4.3 PROOF OF COROLLARY 1 (C)

Suppose $\Sigma \Pi_{\text{std}}^\perp \theta^\star = \lambda \Pi_{\text{std}}^\perp \theta^\star$ for some $\lambda_{\min}(\Sigma) \leq \lambda \leq \lambda_{\max}(\Sigma)$. Then starting with the expression (15),

$$\frac{x_{\text{ext}}^\top \Pi_{\text{std}}^\perp \Sigma \Pi_{\text{std}}^\perp x_{\text{ext}}}{x_{\text{ext}}^\top \Pi_{\text{std}}^\perp x_{\text{ext}}} - 2\frac{(\Pi_{\text{std}}^\perp x_{\text{ext}})^\top \Sigma \Pi_{\text{std}}^\perp \theta^\star}{x_{\text{ext}}^\top \Pi_{\text{std}}^\perp \theta^\star} = \frac{x_{\text{ext}}^\top \Pi_{\text{std}}^\perp \Sigma \Pi_{\text{std}}^\perp x_{\text{ext}}}{x_{\text{ext}}^\top \Pi_{\text{std}}^\perp x_{\text{ext}}} - 2\lambda$$

$$\leq \lambda_{\max}(\Sigma) - 2\lambda < 0$$

by applying $\frac{\lambda_{\max}(\Sigma)}{\lambda_{\min}(\Sigma)} \leq 2$. Thus when $\Pi_{\text{std}}^\perp \theta^\star$ is an eigenvector of $\Sigma$, there are no augmentations $x_{\text{ext}}$ that increase the bias.

## C VARIANCE OF MINIMUM NORM INTERPOLANTS

In this section, we consider the case where the noise is non-zero, and compute the variances of the two estimators of interest: the standard estimator $\hat{\theta}_{\text{std}}$ and data augmented estimator $\hat{\theta}_{\text{aug}}$. The following theorem provides a general characterization of the relation between variance of the standard estimator and variance of the augmented estimator. Theorem 2 is a corollary of this general result that we present first.

**Theorem 4** (Variance). *The difference in the variances of a standard and augmented estimator can be expressed as follows:*

$$\frac{1}{\sigma^2}(V(\hat{\theta}_{aug}) - V(\hat{\theta}_{std})) = \underbrace{\text{tr}\left(\Sigma \overline{X}_{ext}^\dagger (\overline{X}_{ext}^\dagger)^\top\right)}_{T_1:\ \text{Variance increase}} - \underbrace{\text{tr}\left(\Sigma \Sigma_{std}^\dagger X_{ext}^\top (I + X_{ext}\Sigma_{std}^\dagger X_{ext}^\top)^{-1} X_{ext}\Sigma_{std}^\dagger\right)}_{T_2:\ \text{Variance reduction}}, \quad (17)$$

*where $\overline{X}_{ext} \stackrel{\text{def}}{=} \Pi_{std}^\perp X_{ext}$, is the component of $X_{ext}$ in the null space of $\Sigma_{std}$.*

*Proof.* Recall from (3) that the $V(\hat{\theta}) = \text{tr}(\text{Cov}(\hat{\theta} \mid X_{\text{std}} X_{\text{ext}})\Sigma)$. For the minimum norm interpolation estimators $\hat{\theta}_{\text{std}}$ and $\hat{\theta}_{\text{aug}}$ in Equation 2, we have the following expressions for the variances of the estimators.

$$V(\hat{\theta}_{\text{std}}) = \sigma^2 \text{tr}\left(\Sigma_{\text{std}}^\dagger \Sigma\right),$$

$$V(\hat{\theta}_{\text{aug}}) = \sigma^2 \text{tr}\left(\Sigma_{\text{aug}}^\dagger \Sigma\right),$$

where $\Sigma_{\text{std}} = X_{\text{std}}^\top X_{\text{std}}$ and $\Sigma_{\text{aug}} = X_{\text{std}}^\top X_{\text{std}} + X_{\text{ext}}^\top X_{\text{ext}}$. Note that since $\Sigma_{\text{std}}, \Sigma_{\text{aug}}$ are unnormalized, the quantities $\Sigma_{\text{std}}^\dagger$ and $\Sigma_{\text{aug}}^\dagger$ decay with $n$ with the variances $V(\hat{\theta}_{\text{std}}), V(\hat{\theta}_{\text{aug}})$ also decay with $n$ as expected. In order to compare $V(\hat{\theta}_{\text{std}})$ and $V(\hat{\theta}_{\text{aug}})$, we need to compare $\Sigma_{\text{std}}^\dagger$ and $\Sigma_{\text{aug}}^\dagger$. Iorder to do this, we leverage the result from Kovanic (1979) on the pseudo-inverse of the sum of two symmetric matrices:

$$(\Sigma_{\text{std}} + X_{\text{ext}}^\top X_{\text{ext}})^\dagger = \Sigma_{\text{std}}^\dagger - \Sigma_{\text{std}}^\dagger X_{\text{ext}}^\top (I + X_{\text{ext}}\Sigma_{\text{std}}^\dagger X_{\text{ext}}^\top)^{-1} X_{\text{ext}}\Sigma_{\text{std}}^\dagger + \overline{X}_{\text{ext}}^\dagger (\overline{X}_{\text{ext}}^\dagger)^\top,$$

where recall that $\overline{X}_{\text{ext}}$ is the component of $X_{\text{ext}}$ in the null space of $\Sigma_{\text{std}}$. Multiplying each term by $\Sigma$ and using linearity of trace, we get the required expression. $\square$

**Proof of Theorem 2**. Theorem 2 follows directly from the general result above (Theorem 4). Note that the terms $T_1$ and $T_2$ are traces of PSD matrices and hence non-negative, and capture the magnitude of variance increase and variance reduction respectively. From Theorem 4, we see that (i) if $X_{\text{ext}}$ is entirely in the span of $\Sigma_{\text{std}}$ making $\overline{X}_{\text{ext}} = 0$, $T_1 = 0$ making $V(\hat{\theta}_{\text{aug}}) \leq V(\hat{\theta}_{\text{std}})$ (ii) On the other extreme, if $X_{\text{ext}}$ is entirely in the null space with $\Sigma_{\text{std}}^{\dagger} X_{\text{ext}} = 0$, $T_2 = 0$ and hence $V(\hat{\theta}_{\text{aug}}) \geq V(\hat{\theta}_{\text{std}})$.

# D   DETAILS FOR SPLINE STAIRCASE

We describe the data distribution, augmentations, and model details for the spline experiment in Figure 4 and toy scenario in Figure 1. Finally, we show that we can construct a simplified family of spline problems where the ratio between test errors of the augmented and standard estimators increases unboundedly as the number of stairs.

## D.1   TRUE MODEL

We consider a finite input domain

$$\mathcal{T} = \{0, \epsilon, 1, 1+\epsilon, ..., s-1, s-1+\epsilon\} \tag{18}$$

for some integer $s$ corresponding to the total number of "stairs" in the staircase problem. Let $\mathcal{T}_{\text{line}} \subset \mathcal{T} = \{0, 1, ..., s-1\}$. We define the underlying function $f^{\star} : \mathbb{R} \mapsto \mathbb{R}$ as $f^{\star}(t) = \lfloor t \rfloor$. This function takes a staircase shape, and is linear when restricted to $\mathcal{T}_{\text{line}}$.

**Sampling training data $X_{\text{std}}$** We describe the data distribution in terms of the one-dimensional input $t$, and by the one-to-one correspondence with spline basis features $x = X(t)$, this also defines the distribution of spline features $x \in \mathcal{X}$. Let $w \in \Delta_s$ define a distribution over $\mathcal{T}_{\text{line}}$ where $\Delta_s$ is the probability simplex of dimension $s$. We define the data distribution with the following generative process for one sample $t$. First, sample a point $i$ from $\mathcal{T}_{\text{line}}$ according to the categorical distribution described by $w$, such that $i \sim \text{Categorical}(w)$. Second, sample $t$ by perturbing $i$ with probability $\delta$ such that

$$t = \begin{cases} i & \text{w.p. } 1-\delta \\ i+\epsilon & \text{w.p. } \delta. \end{cases}$$

The sampled $t$ is in $\mathcal{T}_{\text{line}}$ with probability $1-\delta$ and $\mathcal{T}_{\text{line}}^c$ with probability $\delta$, where we choose $\delta$ to be small.

**Sampling augmented points $X_{\text{ext}}$** For each element $t_i$ in the training set, we augment with $\tilde{T}_i = [\tilde{u} \overset{u.a.r}{\sim} B(t_i)]$, an input chosen uniformly at random from $B(t_i) = \{\lfloor t_i \rfloor, \lfloor t_i \rfloor + \epsilon\}$. Recall that in our work, we consider data augmentation where the targets associated with the augmented points are from the ground truth oracle. Notice that by definition, $f^{\star}(\tilde{t}_i) = f^{\star}(t_i)$ for all $\tilde{t} \in B(t_i)$, and thus we can set the augmented targets to be $\tilde{y}_i = y_i$. This is similar to random data augmentation in images (Yaeger et al., 1996; Krizhevsky et al., 2012), where inputs are perturbed in a way that preserves the label.

## D.2   SPLINE MODEL

We parameterize the spline predictors as $f_{\theta}(t) = \theta^{\top} X(t)$ where $X : \mathbb{R} \to \mathbb{R}^d$ is the cubic B-spline feature mapping (Friedman et al., 2001) and the norm of $f_{\theta}(t)$ can be expressed as $\theta^{\top} M \theta$ for a matrix $M$ that penalizes a large second derivative norm where $[M]_{ij} = \int X_i''(u) X_j''(u) \mathrm{u}$. Notice that the splines problem is a linear regression problem from $\mathbb{R}^d$ to $\mathbb{R}$ in the feature domain $X(t)$, allowing direct application of Theorem 1. As a linear regression problem, we define the finite domain as $\mathcal{X} = \{X(t) : t \in \mathcal{T}\}$ containing $2s$ elements in $\mathbb{R}^d$. There is a one-to-one correspondence between $t$ and $X(t)$, such that $X^{-1}$ is well-defined. We define the features that correspond to inputs in $\mathcal{T}_{\text{line}}$ as $\mathcal{X}_{\text{line}} = \{x : X^{-1}(x) \in \mathcal{T}_{\text{line}}\}$. Using this feature mapping, there exists a $\theta^{\star}$ such that $f_{\theta^{\star}}(t) = f^{\star}(t)$ for $t \in \mathcal{T}$.

Our hypothesis class is the family of cubic B-splines as defined in (Friedman et al., 2001). Cubic B-splines are piecewise cubic functions, where the endpoints of each cubic function are called the knots. In our example, we fix the knots to be $[0, \epsilon, 1, ..., s-1, s-1+\epsilon]$, which places a knot on every point in $\mathcal{T}$. This ensures that the function class contains an interpolating function on all $t \in \mathcal{T}$, i.e. for some $\theta^{\star}$,

$$f_{\theta^{\star}}(t) = {\theta^{\star}}^{\top} X(t) = f^{\star}(t) = \lfloor t \rfloor.$$

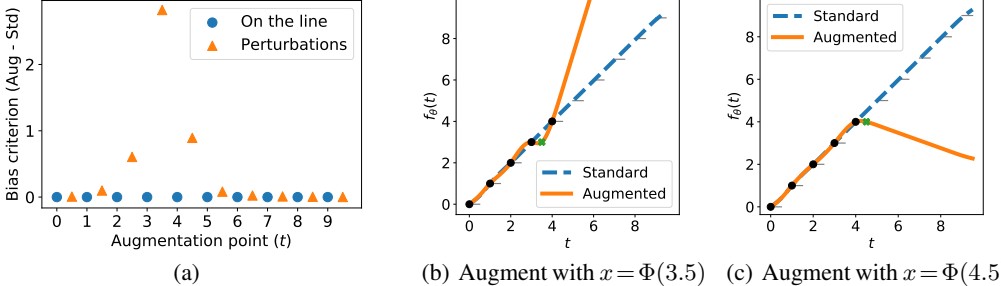

(a)                                    (b) Augment with $x = \Phi(3.5)$    (c) Augment with $x = \Phi(4.5)$

Figure 5: Visualization of the effect of single augmentation points in the noiseless spline problem given an initial dataset $X_{\mathrm{std}} = \{\Phi(t) : t \in \{0,1,2,3,4\}\}$. The standard estimator defined by $X_{\mathrm{std}}$ is linear. **(a)** Plot of the difference term in Corollary 1 (a), is positive when augmenting a single point causes higher test error. Augmenting with points on $\mathcal{X}_{\mathrm{line}}$ does not affect the bias, but augmenting with any element of $\{X(t) : t \in \{2.5, 3.5, 4.5\}\}$ hurts the bias of the augmented estimator dramatically. **(b)**, **(c)** Augmenting with $X(3.5)$ or $X(4.5)$ hurts the bias by changing the direction of extrapolation.

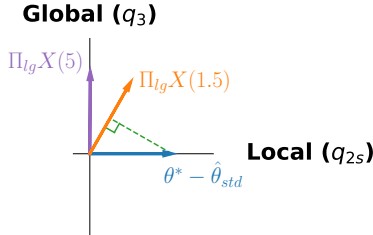

(a) Nullspace projections onto local vs. global

Figure 6: Nullspace projections onto global direction $q_3$ and local direction $q_{2s}$ in $\mathrm{Null}(\Sigma)$ via $\Pi_{\mathrm{lg}}$, representing global and local eigenvectors respectively. The local perturbation $\Pi_{\mathrm{lg}}\hat{\Phi}(1.5)$ has both local and global components, creating a high-error component in the global direction.

We solve the minimum norm problem

$$\hat{\theta}_{\mathrm{std}} = \underset{\theta}{\mathrm{argmin}}\{\theta^\top M \theta : X_{\mathrm{std}}\theta = y_{\mathrm{std}}\} \tag{19}$$

for the standard estimator and the corresponding augmented problem to obtain the augmented estimator.

### D.3 EVALUATING COROLLARY 1 (A) FOR SPLINES

We now illustrate the characterization for the effect of augmentation with different single points in Theorem 1 (a) on the splines problem. We assume the domain to $\mathcal{T}$ as defined in equation 18 with $s = 10$ and our training data to be $X_{\mathrm{std}} = \{X(t) : t \in \{0,1,2,3,4\}\}$. Let *local* perturbations be spline features for $\tilde{t} \notin \mathcal{T}_{\mathrm{line}}$ where $\tilde{t} = t + \epsilon$ is $\epsilon$ away from some $t \in \{0,1,2,3,4\}$ from the training set. We examine all possible single augmentation points in Figure 5 (a) and plot the calculated predictive test error difference as defined in equation (16). Figure 5 shows that augmenting with an additional point from $\{X(t) : t \in \mathcal{T}_{\mathrm{line}}\}$ does not affect the bias, but adding any perturbation point in $\{X(\tilde{t}) : \tilde{t} \in \{2.5, 3.5, 4.5\}\}$ where $\tilde{t} \notin \mathcal{T}_{\mathrm{line}}$ increases the error significantly by changing the direction in which the estimator extrapolates. Particularly, *local* augmentations near the boundary of the original dataset hurt the most while other augmentations do not significantly affect the bias of the augmented estimator.

### D.3.1 LOCAL AND GLOBAL STRUCTURE IN THE SPLINE STAIRCASE

In the spline staircase, the local perturbations can be thought of as fitting high frequency noise in the function space, where fitting them causes a global change in the function.

To see this, we transform the problem to minimum $\ell_2$ norm linear interpolation using features $X_M(t) = X(t)M^{-1/2}$ so that the results from Section 4.1.2 apply directly. Let $\Sigma$ be the population covariance of $X_M$ for a uniform distribution over the discrete domain consisting of $s$ stairs and their perturbations (Figure 1). Let $Q = [q_i]_{i=1}^{2s}$ be the eigenvectors of $\Sigma$ in decreasing order of their corresponding eigenvalues. The visualization in Figure 3 shows that $q_i$ are wave functions in the original input space; the "frequency" of the wave increases as $i$ increases.

Suppose the original training set consists of two points, $X_{std} = [X_M(0), X_M(1)]^\top$. We study the effect of augmenting point $x_{ext}$ in terms of $q_i$ above. First, we find that the first two eigenvectors corresponding to linear functions satisfy $\Pi_{std}^\perp q_1 = \Pi_{std}^\perp q_2 = 0$. Intuitively, this is because the standard estimator is linear. For ease of visualization, we consider the 2D space in $\text{Null}(\Sigma)$ spanned by $\Pi_{std}^\perp q_3$ (global direction, low frequency) and $\Pi_{std}^\perp q_{2s}$ (local direction, high frequency). The matrix $\Pi_{lg} = [\Pi_{std}^\perp q_3, \ \Pi_{std}^\perp q_{2s}]^\top$ projects onto this space. Note that the same results hold when projecting onto all $\Pi_{std}^\perp q_i$ in $\text{Null}(\Sigma)$.

In terms of the simple 3-D example in Section 4.1.1, the global direction corresponds to the costly direction with large eigenvalue, as changes in global structure heavily affect the predictive test error. Figure 6 plots the projections $\Pi_{lg}\theta^\star$ and $\Pi_{lg}X_{ext}$ for different $X_{ext}$. When $\theta^\star$ has high frequency variations and is complex, $\Pi_{lg}\theta^\star = (\theta^\star - \hat{\theta}_{std})$ is aligned with the local dimension. For $x_{ext}$ immediately local to training points, the projection $\Pi_{lg}x_{ext}$ (orange vector in Figure 6) has both local and global components. Augmenting these local perturbations introduces error in the global component. For other $x_{ext}$ farther from training points, $\Pi_{lg}x_{ext}$ (blue vector in Figure 6) is almost entirely global and perpendicular to $\theta^\star - \hat{\theta}_{std}$, leaving bias unchanged. Thus, augmenting data close to original data cause estimators to fit local components at the cost of the costly global component which changes overall structure of the predictor like in Figure 1(middle). The choice of inductive bias in the $M-$norm being minimized results in eigenvectors of $\Sigma$ that correspond to local and global components, dictating this tradeoff.

### D.4 DATA AUGMENTATION CAN BE QUITE PAINFUL FOR SPLINES

We construct a family of spline problems such that as the number the augmented estimator has much higher error than the standard estimator. We assume that our predictors are from the full family of cubic splines.

**Sampling distribution**

We define a modified domain with continuous intervals $\mathcal{T} = \cup_{t=0}^{s-1}[t, t+\epsilon]$. Considering only $s$ which is a multiple of 2, we sample the original data set as described in Section D.1 with the following probability mass $w$:

$$w(t) = \begin{cases} \frac{1-\gamma}{s/2} & t < s/2, t \in \mathcal{T}_{line} \\ \frac{\gamma}{s/2} & t \geq s/2, t \in \mathcal{T}_{line}. \end{cases} \tag{20}$$

for $\gamma \in [0,1)$. We define a probability distribution $P_\mathcal{T}$ on $\mathcal{T}$ for a random variable $T$ by setting $T = Z + S(Z)$ where $Z \sim \text{Categorical}(w)$ and the $Z$-dependent perturbation $S(z)$ is defined as

$$S(z) \sim \begin{cases} \text{Uniform}([z, z+\epsilon]) & \text{w.p. } \delta \\ z, & \text{w.p. } 1-\delta. \end{cases} \tag{21}$$

We obtain the training dataset $X_{std} = \{X(t_1), ..., X(t_n)\}$ by sampling $t_i \sim P_\mathcal{T}$.

**Augmenting with an interval**

Consider a modified augmented estimator for the splines problem, where for each point $t_i$ we augment with the entire interval $[\lfloor t_i \rfloor, \lfloor t_i \rfloor + \epsilon]$ with $\epsilon \in [0, 1/2)$ and the estimator is enforced to output $f_{\hat{\theta}}(x) = y_i = \lfloor t_i \rfloor$ for all $x$ in the interval $[\lfloor t_i \rfloor, \lfloor t_i \rfloor + \epsilon]$. Additionally, suppose that the ratio $s/n = O(1)$ between the number of stairs $s$ and the number of samples $n$ is constant.

In this simplified setting, we can show that the test error of the augmented estimator grows while the test error of the standard estimator decays to 0.

**Theorem 5.** *Let the setting be defined as above. Then with the choice of $\delta = \frac{\log(s^7) - \log(s^7 - 1)}{s}$ and $\gamma = c/s$ for a constant $c \in [0,1)$, the ratio between test errors is lower bounded as*

$$\frac{R(\hat{\theta}_{aug})}{R(\hat{\theta}_{std})} = \Omega(s^2) \tag{22}$$

*which goes to infinity as $s \to \infty$. Furthermore, $R(\hat{\theta}_{std}) \to 0$ as $s \to \infty$.*

*Proof.* We first lower bound the test error of the augmented estimator. Define $E_1$ as the event that only the lower half of the stairs is sampled, i.e. $\{t : t < s/2\}$, which occurs with probability $(1-\gamma)^n$. Let $t^\star = \max_i \lfloor t_i \rfloor$ be the largest "stair" value seen in the training set. Note that the min-norm augmented estimator will extrapolate with zero derivative for $t \geq \max_i \lfloor t_i \rfloor$. This is because on the interval $[t^\star, t^\star + \epsilon]$, the augmented estimator is forced to have zero derivative, and the solution minimizing the second derivative of the prediction continues with zero derivative for all $t \geq t^\star$. In the event $E_1$, $t^\star \leq s/2 - 1$, where $t^* = s/2 - 1$ achieves the lowest error in this event. As a result, on the points in the second half of the staircase, i.e. $t = \{t \in \mathcal{T} : t > \frac{s}{2} - 1\}$, the augmented estimator incurs large error:

$$R(\hat{\theta}_{\text{aug}} \mid E_1) \geq \sum_{t=s/2}^{s} (t - (s/2 - 1))^2 \cdot \frac{\gamma}{s/2}$$

$$= \sum_{t=1}^{s/2} t^2 \cdot \frac{\gamma}{s/2} = \frac{\gamma}{6}(s^2 + 2s + 1).$$

Therefore the expected risk of the augmented estimator is bounded by

$$R(\hat{\theta}_{\text{aug}}) \geq R(\hat{\theta}_{\text{aug}} \mid E_1) P(E_1) = \frac{\gamma}{6}(s^2 + 2s + 1)(1 - \gamma)^n$$

$$\geq \frac{1}{6}\gamma(1 - \gamma n)(s^2 + 2s + 1)$$

$$= \Omega\left(\frac{c - c^2}{s}(s^2 + 2s + 1)\right) = \Omega(s)$$

where in the first line, we note that the error on each interval is the same and the probability of each interval is $(1-\delta)\frac{\gamma}{s/2} + \epsilon \frac{\delta}{\epsilon} \cdot \frac{\gamma}{s/2} = \frac{\gamma}{s/2}$.

Next we upper bound the test error of the standard estimator. Define $E_2$ to be the event where all points are sampled from $\mathcal{T}_{\text{line}}$, which occurs with probability $(1 - \delta)^n$. In this case, the standard estimator is linear and fits the points on $\mathcal{T}_{\text{line}}$ with zero error, while incurring error for all points not in $\mathcal{T}_{\text{line}}$. Note that the probability density of sampling a point not in $\mathcal{T}_{\text{line}}$ is either $\frac{\delta}{\epsilon} \cdot \frac{1-\gamma}{s/2}$ or $\frac{\delta}{\epsilon} \cdot \frac{\gamma}{s/2}$, which we upper bound as $\frac{\delta}{\epsilon} \cdot \frac{1}{s/2}$.

$$R(\hat{\theta}_{\text{std}} \mid E_2) = \sum_{t=1}^{s-1} \frac{\delta}{\epsilon} \cdot \frac{1}{s/2} \int_0^{\epsilon} u^2 \, du = \frac{\delta}{\epsilon} \cdot \frac{1}{s/2} O(s\epsilon^3)$$

$$= O(\delta)$$

Therefore for event $E_2$, the expected error is bounded as

$$R(\hat{\theta}_{\text{std}} \mid E_2) P(E_2) = O(\delta)(1 - \delta)^n$$

$$= O(\delta)e^{-\delta n}$$

$$= O\left(\delta \cdot \frac{s^7 - 1}{s^7}\right)$$

$$= O(\delta) = O\left(\frac{\log(s^7) - \log(s^7 - 1)}{s}\right) = O(1/s)$$

since $\log(s^7) - \log(s^7 - 1) \leq 1$ for $s \geq 2$. For the complementary event $E_2^c$, note that cubic spline predictors can grow only as $O(t^3)$, with error $O(t^6)$. Therefore the expected error for case $E_2^c$ is bounded as

$$R(\hat{\theta}_{\text{std}} \mid E_2^c) P(E_2^c) \leq O(t^6)(1 - e^{-\delta n})$$

$$= O(t^6) O\left(\frac{1}{s^7}\right) = O(1/s)$$

Putting the parts together yields

$$R(\hat{\theta}_{\text{std}}) = R(\hat{\theta}_{\text{std}} \mid E_2)P(E_2) + R(\hat{\theta}_{\text{std}} \mid E_2^c)P(E_2^c)$$
$$\leq O(1/s) + O(1/s) = O(1/s).$$

Thus overall, $R(\hat{\theta}_{\text{std}}) = O(1/s)$ and combining the bounds yields the result. $\square$

## E    EFFECT OF SAMPLE SIZE ON ERROR INCREASE VIA AUGMENTATION

In this section, we discuss what our analysis of the effect of augmentation on the error of mininum interpolants says about the trends with respect to varying sample sizes of the standard training set $(X_{\text{std}}, y_{\text{std}})$.

**Trends in variance.**  We refer the reader to the precise expression for the difference in variance provided in Theorem 4. Let us consider a case where data augmentation causes an increase in variance. For simplicity, $X_{\text{ext}} \perp X_{\text{std}}$, across all sample sizes in the small sample regime. For a fixed $X_{\text{ext}}$, we see that the magnitude of variance increase is governed by $\Pi_{\text{std}}^{\perp}X_{\text{ext}}$ which decreases as we get more standard training points.

**Trends in bias.**  Recall the expressions for $B(\hat{\theta}_{\text{std}})$ and $B(\hat{\theta}_{\text{aug}})$ from Equation 4. Using the same notation as that of Theorem 5, we have the following expression for the amount of bias increase. Let $v = \Pi_{\text{std}}^{\perp}\Pi_{\text{aug}}\theta^{\star}$ and $v = \Pi_{\text{aug}}^{\perp}\theta^{\star}$. We have,

$$B(\hat{\theta}_{\text{aug}}) - B(\hat{\theta}_{\text{std}}) = -v^{\top}\Sigma v - 2w^{\top}\Sigma v, \qquad (23)$$

where $w^{\top}\Sigma v$ is a negative quantity when data augmentation causes an increase in bias. Recall that we are in the small sample regime where $\Sigma_{\text{std}}$ is not invertible for a range of sample sizes. Suppose we augment with $X_{\text{ext}}$ such that $X_{\text{ext}} \in \text{Null}(\Sigma_{\text{std}})$ in this regime of interest. In this case, we can write $v = \Pi_{\text{std}}^{\perp}\Pi_{\text{aug}}^{\perp}\theta^{\star} = u^{\top}\theta^{\star}$. For a fixed problem setting $\theta^{\star}$, we see that $v$ is fixed. Let us now look at $w = \Pi_{\text{aug}}^{\perp}\theta^{\star}$. Recall that $\Pi_{\text{aug}}^{\perp}$ is the projection matrix onto $\Sigma_{\text{std}} + X_{\text{ext}}^{\top}X_{\text{ext}}$. For a fixed $X_{\text{ext}}$, as the null space of $\Sigma_{\text{std}}$ shrinks with more training points, $w$ decreases. Note that the magnitude of increase in bias decreases as $w$ decreases (for a fixed $v$). This suggests that the effect of augmentation on bias should decrease as we get more samples, in the small data regime.

Our heuristic calculations in some settings of $X_{\text{ext}}$ in minimum norm interpolation in linear regression suggest that the overall increase in test error should decrease as we increase the sample size of the original training set. Empirically, we find similar trends when performing adversarial data augmentations with varying training set sizes.

## F    X-REGULARIZATION

### F.1    X-REGULARIZATION FOR LINEAR REGRESSION

In this section, we prove Theorem 3, which we reproduce here.

**Theorem 3.**  *Assume the noiseless linear model $y = x^{\top}\theta^{\star}$. Let $\theta_{\text{int-std}}$ be an arbitrary interpolant of the standard data, i.e. $X_{\text{std}}\theta_{\text{int-std}} = y_{\text{std}}$. Let $\hat{\theta}_{\text{x-aug}}$ be the X-regularized interpolant* (6). *Then*

$$R\big(\hat{\theta}_{\text{x-aug}}\big) \leq R(\theta_{\text{int-std}}).$$

*Proof.*  Let $\{u_i\}$ be an orthonormal basis of the kernel $\text{Null}(\Sigma_{\text{std}} + X_{\text{ext}}^{\top}X_{\text{ext}})$ and $\{v_i\}$ be an orthonormal basis for $\text{Null}(\Sigma_{\text{std}}) \setminus \text{span}(\{u_i\})$. Let $U$ and $V$ be the linear operators defined by $Uw = \sum_i u_i w_i$ and $Vw = \sum_i v_i w_i$, respectively, noting that $U^{\top}V = 0$. Defining $\Pi_{\text{std}}^{\perp} := (I - \Sigma_{\text{std}}^{\dagger}\Sigma_{\text{std}})$ to be the projection onto the null space of $X_{\text{std}}$, we see that there are unique vectors $\rho, \alpha$ such that

$$\theta^{\star} = (I - \Pi_{\text{std}}^{\perp})\theta^{\star} + U\rho + V\alpha. \qquad (24a)$$

As $\theta_{\text{int-std}}$ interpolates the standard data, we also have

$$\theta_{\text{int-std}} = (I - \Pi_{\text{std}}^{\perp})\theta^{\star} + Uw + Vz, \qquad (24b)$$

as $X_{\text{std}}Uw = X_{\text{std}}Vz = 0$, and finally,

$$\hat{\theta}_{\text{x-aug}} = (I - \Pi_{\text{std}}^{\perp})\theta^{\star} + U\rho + V\lambda \tag{24c}$$

where we note the common $\rho$ between Eqs. (24a) and (24c).

Using the representations (24) we may provide an alternative formulation for the augmented estimator (6), using this to prove the theorem. Indeed, writing $\theta_{\text{int-std}} - \hat{\theta}_{\text{x-aug}} = U(w - \rho) + V(z - \lambda)$, we immediately have that the estimator has the form (24c), with the choice

$$\lambda = \underset{\lambda}{\operatorname{argmin}}\left\{(U(w - \rho) + V(z - \lambda))^{\top}\Sigma(U(w - \rho) + V(z - \lambda))\right\}.$$

The optimality conditions for this quadratic imply that

$$V^{\top}\Sigma V(\lambda - z) = V^{\top}\Sigma U(w - \rho). \tag{25}$$

Now, recall that the predictive test error of a vector $\theta$ is $R(\theta) = (\theta - \theta^{\star})^{\top}\Sigma(\theta - \theta^{\star}) = \|\theta - \theta^{\star}\|_{\Sigma}^{2}$, using Mahalanobis norm notation. In particular, a few quadratic expansions yield

$$R(\theta_{\text{int-std}}) - R(\hat{\theta}_{\text{x-aug}})$$
$$= \|U(w - \rho) + V(z - \alpha)\|_{\Sigma}^{2} - \|V(\lambda - \alpha)\|_{\Sigma}^{2}$$
$$= \|U(w - \rho) + Vz\|_{\Sigma}^{2} + \|V\alpha\|_{\Sigma}^{2} - 2(U(w - \rho) + Vz)^{\top}\Sigma V\alpha - \|V\lambda\|_{\Sigma}^{2} - \|V\alpha\|_{\Sigma}^{2} + 2(V\lambda)^{\top}\Sigma V\alpha$$
$$\overset{(i)}{=} \|U(w - \rho) + Vz\|_{\Sigma}^{2} - 2(V\lambda)^{\top}\Sigma V\alpha - \|V\lambda\|_{\Sigma}^{2} + 2(V\lambda)^{\top}V\alpha$$
$$= \|U(w - \rho) + Vz\|_{\Sigma}^{2} - \|V\lambda\|_{\Sigma}^{2}, \tag{26}$$

where step $(i)$ used that $(U(w - \rho))^{\top}\Sigma V = (V(\lambda - z))^{\top}\Sigma V$ from the optimality conditions (25).

Finally, we consider the rightmost term in equality (26). Again using the optimality conditions (25), we have

$$\|V\lambda\|_{\Sigma}^{2} = \lambda^{\top}V^{\top}\Sigma^{1/2}\Sigma^{1/2}(U(w - \rho) + Vz) \le \|V\lambda\|_{\Sigma}\|U(w - \rho) + Vz\|_{\Sigma}$$

by Cauchy-Schwarz. Revisiting equality (26), we obtain

$$R(\theta_{\text{int-std}}) - R(\hat{\theta}_{\text{x-aug}}) = \|U(w - \rho) + Vz\|_{\Sigma}^{2} - \frac{\|V\lambda\|_{\Sigma}^{4}}{\|V\lambda\|_{\Sigma}^{2}}$$

$$\ge \|U(w - \rho) + Vz\|_{\Sigma}^{2} - \frac{\|V\lambda\|_{\Sigma}^{2}\|U(w - \rho) + Vz\|_{\Sigma}^{2}}{\|V\lambda\|_{\Sigma}^{2}} = 0,$$

as desired. $\qquad\square$

## F.2 X-REGULARIZATION FOR DATA AUGMENTATIONS THAT PROMOTE ROBUSTNESS

The main motivation of our work is to provide a method to perform data augmentation such that the benefits such as robustness are preserved, without seeing the undesirable drop in standard accuracy. The general X-regularized estimator (Equation 7) holds for any form of augmentation. We now write out the exact loss functions when we apply X-regularization to two forms of adversarial training: Projected Gradient Adversarial Training of (Madry et al., 2018) and TRADES (Zhang et al., 2019). Throughout, we assume the same notation as that used in the definition of the general estimator. $X_{\text{std}}, y_{\text{std}}$ denote the standard training set and we have access to $m$ unlabeled points $\tilde{x}_i, i = 1, ... m$.

### F.2.1 PROJECTED GRADIENT ADVERSARIAL TRAINING

Note that the unlabeled data can be perturbed to obtain more extra data, because of the special structure of the extra points added: every training point generates a perturbed extra training point. This leads to the following natural generalization, where we obtain adversarial perturbations from the unlabeled data, and label them with the pseudo-label generated from the standard trained model. As the distance measure, we use the same loss that is used for classification. Put together, we have

$$\hat{\theta}_{\text{x-aug}} := \underset{\theta}{\operatorname{argmin}}\left\{ N^{-1}\sum_{(x,y)\in[X_{\text{std}}, y_{\text{std}}]}\ell(f_{\theta}(x), y) + \beta\,\ell(f_{\theta}(x_{\text{adv}}), y)\right.$$
$$\left. + \lambda m^{-1}\sum_{i=1}^{m}\ell(f_{\theta}(\tilde{x}_i), f_{\hat{\theta}_{\text{std}}}(\tilde{x}_i)) + \beta\,\ell(f_{\theta}(\tilde{x}_{\text{adv}\,i}), f_{\hat{\theta}_{\text{std}}}(\tilde{x}_i))\right\}, \tag{27}$$

In practice, $x_{\text{adv}}$ is found by performing a few steps of projected gradient method on $\ell(f_\theta(x),y)$, and similarly $\tilde{x}_{\text{adv}}$ by performing a few steps of projected gradient method on $\ell(f_\theta(\tilde{x}),f_{\hat{\theta}_{\text{std}}}(\tilde{x}))$.

### F.2.2 TRADES

TRADES was a modification of the projected gradient adversarial training algorithm of (Madry et al., 2018). Here, the loss function is modified in the following way, instead of operating on the label directly, the robustness term operates on the normalized logits, which can be thought of as probabilities of different labels. Using their modified loss and applying X-regularization leads to the following.

$$\hat{\theta}_{\text{x-aug}} := \underset{\theta}{\text{argmin}} \Bigg\{ N^{-1} \sum_{(x,y)\in[X_{\text{std}},y_{\text{std}}]} \ell(f_\theta(x),y) + \beta\, KL(p_\theta(x_{\text{adv}})||p_\theta(x))$$

$$+ \lambda m^{-1} \sum_{i=1}^{m} \ell(f_\theta(\tilde{x}_i),f_{\hat{\theta}_{\text{std}}}(\tilde{x}_i)) + \beta\, KL(p_\theta(\tilde{x}_{\text{adv}\,i})||p_{\hat{\theta}_{\text{std}}}(\tilde{x}_i)) \Bigg\}, \qquad (28)$$

where $KL(p_\theta(x),p_\theta(x_{\text{adv}}))$ is the KL divergence between the probability over class labels assigned to $x$ and $x_{\text{adv}}$.

## G EXPERIMENTAL DETAILS

### G.1 SPLINE SIMULATIONS

For spline simulations in Figure 1 and Figure 4, we implement the optimization of the standard and robust objectives using the basis described in (Friedman et al., 2001). The penalty matrix $M$ computes second-order finite differences of the parameters $\theta$. We solve the min-norm objective directly using CVXPY (Diamond & Boyd, 2016). Each point in Figure 4(a) represents the average test error over 25 trials of randomly sampled training datasets between 22 and 1000 samples. Shaded regions represent 1 standard deviation.

### G.2 X-REGULARIZATION AS ROBUST SELF-TRAINING

In the adversarial training setting where augmentations are generated through transformations of existing data, it is natural to instantiate X-regularization as robust self-training, as discussed in Section F. We evaluate the performance of X-regularization applied to $\ell_\infty$ adversarial perturbations, adversarial rotations, and random rotations.

### G.2.1 SUBSAMPLING CIFAR-10

We augment with $\ell_\infty$ adversarial perturbations of various sizes. In each epoch, we find the augmented examples via Projected Gradient Ascent on the multiclass logistic loss (cross-entropy loss) of the incorrect class. Training the augmented estimator in this setup uses essentially the adversarial training procedure of (Madry et al., 2018), with equal weight on both the "clean" and adversarial examples during training.

We instantiate the general X-regularization estimator as robust self-training defined in (27). We compare the test error of the augmented estimator with an estimator trained using RST. We apply RST to adversarial training algorithms in CIFAR-10 using 500k unlabeled examples sourced from Tiny Images, as in (Carmon et al., 2019).

We use Wide ResNet 40-2 models (Zagoruyko & Komodakis, 2016) while varying the number of samples in CIFAR-10. We sub-sample CIFAR-10 by factors of $\{1,2,5,8,10,20,40\}$ in Figure 4(a) and $\{1,2,5,8,10\}$ in Figure 4(b). For sub-sample factors 1 to 20, we report results averaged from 2 trials for each model. For sub-sample factors greater than 20, we average over 5 trials. All models are trained for 200 epochs with respect to the size of the labeled training dataset and all achieve almost 100% standard and robust training accuracy.

We evaluate the robustness of models to the strong PGD-attack with 40 steps and 5 restarts. In Figure 4(b), we used a simple heuristic to set the regularization strength $\lambda$ in Equation (27) to be $\lambda = \min(0.9,\gamma)/(1-\min(0.9,\gamma))$ where $\gamma \in [0,1]$ is the fraction of the original CIFAR-10 dataset

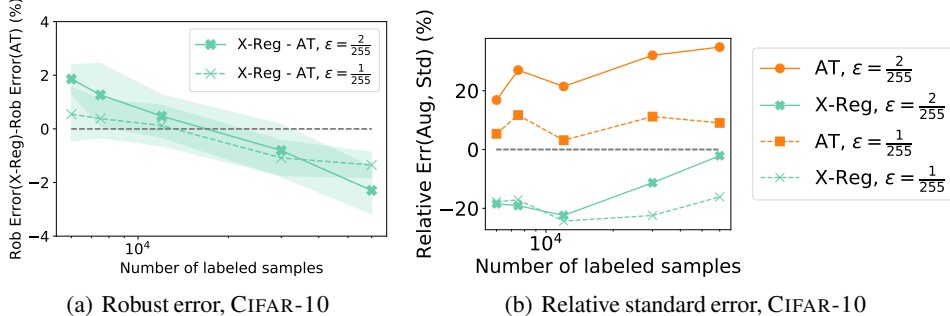

(a) Robust error, CIFAR-10                    (b) Relative standard error, CIFAR-10

Figure 7: **(a)** Difference in robust test error between our X-regularized adversarial training model and the vanilla adversarial training (AT) model for CIFAR-10. X-regularization (instantiated as RST) keeps the robust accuracy within 2% of the AT model for small subsamples and even improves over the AT model for larger subsamples of CIFAR-10. **(b)** Relative difference in standard error between augmented estimators (our X-regularized model and the AT model) and the standard estimator on CIFAR-10. We achieve up to 20% better standard error than the standard model for small subsamples.

|  | Standard | AT | RST+AT |
|---|---|---|---|
| Standard Test Acc | 94.63% | 94.15% | **95.47%** |
| Robust Test Acc ($\epsilon = 1/255$) | - | 85.59% | **87.20%** |

Table 2: Accuracies for the standard, vanilla adversarial training (AT), and AT with X-regularization (RST) for $\epsilon = 1/255$ on the full CIFAR-10 dataset. Accuracies are averaged over two trials. The robust test accuracy of the standard model is near 0%.

|  | Standard | AT | X-reg |
|---|---|---|---|
| Standard Test Acc | 94.63% | 92.69% | **94.86%** |
| Robust Test Acc ($\epsilon = 2/255$) | - | 77.87% | **80.46%** |

Table 3: Accuracies for the standard, vanilla adversarial training (AT), and AT with X-regularization (X-reg) for $\epsilon = 2/255$ on the full CIFAR-10 dataset. Accuracies are averaged over two trials. The robust test accuracy of the standard model is near 0%.

sampled. Intuitively, we give more weight to the unlabeled data when the original dataset is larger, meaning that the standard estimator produces more accurate pseudo-labels. We fix $\beta = 5$.

Figure 7 shows that the robust accuracy of the RST model stays within 2% of the robust model (trained using PGD adversarial training) for all subsamples, and even improves upon the robust model on the full dataset (Tables 2,3).

Note that we cannot directly compare the empirical performance of RST+adversarial training on CIFAR-10 with other methods to obtain robust models that are modifications of vanilla adversarial training. We use a smaller model due to computational constraints enforced by adversarial training. Since the model is small, we could only fit adversarially augmented examples with small $\epsilon = 2/255$, while existing baselines use $\epsilon = 8/255$. Note that even for $\epsilon = 2/255$, adversarial data augmentation leads to an increase in error. We show that RST can fix this. While ensuring models are robust is an important goal in itself, in this work, we view adversarial training through the lens of covariate-shifted data augmentation and study how to use augmented data without increasing test error. We show that X-regularization based methods like RST preserve the other benefits of some kinds of data augmentation like increased robustness to adversarial examples.

### G.2.2 $\ell_\infty$ ADVERSARIAL PERTURBATIONS

In Table 1, we evaluate X-regularization as robust self-training applied to PGD and TRADES adversarial training. The models are trained on the full CIFAR-10 dataset, and models which use

unlabeled data (self-training and X-reg) also use 500k unlabeled examples from Tiny Images. All models except the Interpolated AT and Neural Architecture Search model use the same base model WideResNet 28-10. To evaluate robust accuracy, we use a strong PGD-attack with $40$ steps and $5$ restarts against $\ell_\infty$ perturbations of size $8/255$. For X-regularization models, we set $\lambda = 9$ and $\beta = 5$ in Equation (27) and Equation (28), following the heuristic $\lambda = \min(0.9, \gamma)/(1 - \min(0.9, \gamma))$ for $\gamma = 1$. We train for 200 epochs such that 100% training standard accuracy is attained.

### G.2.3 ADVERSARIAL AND RANDOM ROTATION/TRANSLATIONS

In Table 1 (right), we instantiate X-regularization as robust self-training for adversarial and random rotation/translations, using these transformations as $x_{\text{adv}}$ in Equation (27). The attack model is a grid of rotations of up to 30 degrees and translations of up to $\sim 10\%$ of the image size. The grid consists of 31 linearly spaced rotations and 5 linearly spaced translations in both dimensions. The Worst-of-10 model samples 10 uniformly random transformations of each input and augment with the one where the model performs the worst (causes an incorrect prediction, if it exists). The Random model samples 1 random transformation as the augmented input. All models (besides cited models) use the WRN-40-2 architecture and are trained for 200 epochs. We use the same hyperparameters $\lambda, \beta$ as in G.2.2 for Equation (27).

## H COMPARISON TO STANDARD SELF-TRAINING ALGORITHMS

The main objective of X-regularization is to allow to perform data augmentation without sacrificing standard accuracy. This is done by smoothing an augmented estimator to provide labels close to a standard non-augmented estimator on the unlabeled data. This is closely related to but different two broad kinds of semi-supervised learning.

1. Self-training (pseudo-labeling): Classical self-training does not deal with data augmentation or robustness. We view X-regularization as a a generalization of self-training in the context of data augmentations. Here the pseudolabels are generated by a standard non-augmented estimator that is *not* trained on the labeled augmented points. In contrast, standard self-training would just use all labeled data to generate pseudo-labels. However, since some augmentations cause a drop in standard accuracy, and hence this would generate worse pseudo-labels than the X-regularized version.

2. Consistency based regularization: Another popular semi-supervised learning strategy is based on enforcing consistency in a model's predictions across various perturbations of the unlabeled data (Miyato et al., 2018; Xie et al., 2019; Sajjadi et al., 2016; Laine & Aila, 2017)). X-regularization is similar in spirit, but has an additional crucial component. We generate pseudo-labels first by performing standard training, and rather than enforcing simply consistency across perturbations, we enforce that the unlabeled data and perturbations are matched with the pseudo-labels generated.

