# OpenReview forum: "When Covariate-shifted Data Augmentation Increases Test Error And How to Fix It"
_ICLR.cc/2020/Conference — Reject_

### Official Review · AnonReviewer3 · 2019-10-17
**Official Blind Review #3**

**Rating:** 3

**Review:**

This paper studies data augmentation in the regime where labels for the augmented datapoints are known. Special emphasis is put on the study of overparametrised linear models with minimum Euclidean norm of the regression weights (a.k.a. ridgeless regression). The results of this study are then used to motivate their “X-regularization” method, a semi-supervised learning algorithm which they test on the task of improving accuracy of adversarially trained models. The authors report improvement in accuracy of adversarially trained classifiers on CIFAR-10 when X-regularization is applied.

While I think the paper could be potentially interesting to the ICLR community, I am currently leaning towards rejection for the following two reasons: (i) I am confused by significant chunks of the theoretical derivations;  (ii) the paper is hard  follow at several places, seems to be hastily written, and not well placed in the existing literature. Finally, I would like to state that since the paper is longer than 8 pages, I am holding it to a higher standard than an 8 page one as instructed by the guidelines.


Major comments:

- I am very confused by the way you compute the (uncentred) sample covariance for the augmented dataset. On the penultimate line before eq.5, you state that Sigma_data = 1 / n X^T X if I interpret n as the total number of rows of X (which is IMHO correct). But then a couple of lines below you say that Sigma_aug = 1 / n (X_std^T X_std + alpha X_ext^T X_ext) which by the above logic should have been Sigma_aug = 1 / ((1 + alpha) n) (X_std^T X_std + X_ext^T X_ext) which is different from your definition. Inspecting appendix A.1, you also use Sigma_data = Sigma_std + alpha Sigma_ext whereas the above would suggest Sigma_data = 1 / ((1 + alpha)n) (n Sigma_data + alpha n Sigma_ext) = 1 / (1 + alpha) (Sigma_data + alpha Sigma_ext)?! Can you please clarify? Note that I did not go through the rest of the proof in the appendix.

- On a related note, I am also rather confused by your example in sect.3.1.1. In particular, if you observe e_1, e_2, and e_3, and you are in the scenario where you can get the true labels for the X_ext data points + there is no noise on y (sigma = 0), then you will get theta_aug = (X^T X)^dagger X^T X theta* = Id^(-1) Id theta* = theta* (where Id stands for the identity matrix), i.e., theta_aug = theta* and so there will be zero generalisation error since y = x^T theta* is the correct data generating process in this scenario. However, in the first display on p.4, you claim the bias (and thus generalisation error) of theta_aug is non-zero. The rest of the section similarly seems to contradict the theta_aug = theta*. Can you please clarify?

- Since the rest of sect.3 also uses the Sigma_aug = Sigma_data + alpha Sigma_ext instead of scaling this expression by Sigma_aug = 1 / (1 + alpha) (Sigma_data + alpha Sigma_ext), I will withhold my judgement until you could please clarify this confusion.

- Can you please point me to where I can find more detail about fig.1? In particular, how was the right plot produced (incl. how many repetitions of the experiment were executed)? Also, is it true that the green points in the middle plot were not chosen at random? If so, this should be clarified in the caption and the text.

- Since all of your non-toy experiments are focused on adversarial learning, it might benefit the paper to advertise this focus more clearly from page one. Furthermore, have you tested X-regularization with non-adversarial data augmentation (on CIFAR-10 or other standard benchmark) please?

- While you briefly mention the existence of alternatives to your algorithm, you don’t include any of these in your experiments section. I would suggest including some benchmark results (even if this only means restating numbers from other papers with citation).


Minor comments:

- In the abstract, you claim about X-regularization: “We prove that our new estimator never increases test error.” while sect.4 you say that this guarantee only holds for linear models. I think this should be clarified in the abstract.

- p.1 in par.1, you say “but in practice, we often work with models which can fit the augmented training data perfectly” (similar statements also appear throughout the paper, e.g., the last sentence of par.3 on p.1). I am confused by this statement because many augmentation schemes have full support in the input space, i.e., you will eventually have to be able to classify correctly any point in the input space. Are you assuming that the conditional distribution of y given x is almost surely deterministic, and using some universal approximation theorem for neural networks (even those tend to have some restrictions on the y = f(x) function!)? Please clarify.

- p.1 in par.3, “... may increase increase error …” -> “may increase error”

- Second to last paragraph on p.2, “Y = X theta + sigma N(0, I)” is mixing random variables and distributions. Please consider replacing “N(0, I)” with “epsilon, epsilon ~ N(0, I)“. Also, comparing with eq.1, did you mean “sigma^2” in eq.1?

- Last paragraph on p.2, “compare the performance two estimators” -> “ compare the performance of two estimators”.

- Top of p.2 in eq.3: the definition of the augmented estimator is somewhat confusing. If the augmentation causes the total number of points (i.e., n + \alpha n) to be higher than d (the number of parameters), then the set on the right hand side can be empty (and will be unless there is some perfect multicollinearity). Can you please clarify how is the augmented estimator defined in that case?

- Please reconsider use of contractions (e.g., “doesn’t” -> “does not” in par.2 of sect.3) throughout the paper.

- In eq.5, in the expression for bias, I think you are missing star superscript for the 2nd theta.

- Under eq.5, “... we say that augmentation is safe if the predictive risk or bias does not increase, and hurtful if it does.” Can you please clarify whether the use of “or” is in the strictly logical sense, i.e., arbitrarily large increase in variance is fine as long as bias does not increase (in which case the second part of the statement and thus the full statement is still true)?

- Beginning of “Large Sample Regime” paragraph of sect.3, you claim  “Common intuition from a statistical standpoint would suggest that more data is always good.” I am not sure I agree: there is a plethora of literature on data poisoning, robustness, differential privacy, adversarial examples, etc., which shows that adding data can significantly worsen performance of models. Even without reading these papers, it seems quite natural that adding **arbitrary** data into the dataset may not be the best idea (even in the large sample regime).

- Unresolved equation reference on p.11, 2nd line. Also, in the very next sentence, “yields zero and variance …” -> “yields zero bias and variance ...“?! Broken references also on p.14 (twice).

**Experience Assessment:**

I have read many papers in this area.

**Review Assessment: Checking Correctness Of Derivations And Theory:**

I assessed the sensibility of the derivations and theory.

**Review Assessment: Checking Correctness Of Experiments:**

I did not assess the experiments.

**Review Assessment: Thoroughness In Paper Reading:**

I read the paper at least twice and used my best judgement in assessing the paper.

---

> ### Author Response · Authors · 2019-11-15
> **Author Response to R3**
>
> We thank R3 for their feedback, questions and suggestions. We have significantly improved our presentation of both the theoretical results and the contributions and positioning. We request R3 to refer to the general comment above and evaluate the revised paper.
>
> Addressing the major comments,
> - “Sample covariance of augmented dataset”:  In the previous version, we used \Sigma_std to denote empirical covariance of standard points and \Sigma_ext for the empirical covariance of additional training points. \Sigma_aug did not correspond to the empirical covariance of augmented data. We defined \Sigma_aug to have a scaling of n in the denominator because we wanted to analytically compare (as a function of n) what happens with n points from the standard distribution with and without the extra \alpha n points. We note that these are just notational issues and do not affect the correctness of the results. However, as R3 points out a possible confusion in notation, we changed the presentation in our revision to work with only the unnormalized quantities, thereby resolving the confusion about what is the right normalization quantity. In our revision, \Sigma_std = X_std^\top X_std; \Sigma_ext = X_ext^\top X_ext; \Sigma_aug = \Sigma_std + \Sigma_ext.
>
> - “x_ext = [e_1; e_2] and hence bias upon augmentation in the 3D example”: Unfortunately, we had a typographical error in our description. In our simple 3D example, the augmented point should be the single point x_ext = e1 + e2, such that the nullspace of the original+extra data is nonzero and the augmented estimator is still biased. The picture and following text reflect the correct augmentation (e1+e2). We apologize for this typographical error and thank R3 for pointing it out. We have fixed this in our revision.
>
> - “Sigma_aug”: We have changed notation to make this clearer. We note that all our results only depend on the column spaces and null spaces of \Sigma_aug and \Sigma_std. The scaling does not change anything. However, we do acknowledge the source of confusion and have revised accordingly.
>
> - “Clarifications about Fig. 1”: We have now moved the plot in Figure 1 (right) to Figure 4. The spline simulation details are present in Appendix G. Each point in the spline subsampling experiment is an average of 25 trials. In the middle pane of Fig. 1, every training point has an accompanying green point that is a local transformation. In general, we allow arbitrary augmentations in our framework, including local perturbations, and we clarify the augmentations which fall in the framework in the revision.
>
> - “Focus on adversarial training from the intro”: We have changed our introduction and presentation more generally to portray adversarial training as the motivation of our study
>
> - “Baseline comparisons”: We have added more baseline comparisons and different perturbations - see the general comment (d) for more details.
>
> We also thank R3 for carefully pointing out minor comments. We note that we have fixed all these in the revision.

---

### Official Review · AnonReviewer1 · 2019-10-25
**Official Blind Review #1**

**Rating:** 6

**Review:**

Summary: This paper provides some theory into the question of whether data augmentation can hurt test-set performance. To paraphrase the theory, data augmentation can hurt when it causes the model to learn a spurious local details instead of global structure, even if the augmented data comes from the same (predictive) distribution, and even if the true model lies in the hypothesis class of the learned model. Ironically these effects may be diminished in the large-sample regime, where data augmentation is less important in practice. Motivated by these issues, the authors propose "X-regularization" which requires that models trained on standard and augmented data produce similar predictions on unlabeled data. The paper includes a few experiments on a toy staircase regression problem as well as some ResNet experiments on CIFAR-10.

Review: Overall I think this paper provides an interesting perspective on when and why augmented data can increase test error. Most of the theory deals with a simple convex problem, but some work is done to measure whether the insights carry over to more realistic settings. I think the main ways the paper could be improved are a) better explanation of connections between the theoretical assumptions and practice and b) better treatment of "X-regularization". For a) in particular, more discussion of which exiting data augmentation schemes fall under the theoretical framework would be useful. For b), X-regularization seems tacked on to the end of the paper, despite the potential strength of the contribution. It is not compared directly to existing semi-supervised or robust learning methods. As it currently stands I think the paper spends a bit too much time in the main text on the toy problem and theory, and could be bolstered by additional discussion of x-regularization. But, I think it's acceptable in its current form.

Specific comments:
- In Figure 1 the green crosses use such wide lines that they look like dots at first glance. When the text referred to "crosses" I was confused. I'd suggest making them more obviously crosses.
- "the distribution of Px of the augmented dataset is potentially different than Stairs on the original dataset" Up until this point, your setting was not specific to the "Stairs" problem so I'm not sure why you're mentioning it here. This sentence would make more sense to me if you removed the words "Stairs on the".
- I think it would be useful if you provided some examples of data augmentation schemes (that are used in practice) that satisfy and do not satisfy the assumptions of your theoretical analysis.
- I would also suggest that you provide some additional connections or discussion of the relation between the behavior of your setting (section 2) to the behavior of models we actually use in practice (neural nets). How much of the behavior do we expect to carry over? Why?
- "AT augments with imperceptible perturbations of training images with the corresponding correct target and thereby falls in our framework of covariate-shifted data augmentation." You should mention the assumptions implicit in this definition, namely that perturbing an image with a perturbation whose l_inf norm is less than some threshold will not change the target class. This is what connects it to your theoretical setting and so is an important detail.
- Ma et al. (2018); Belkin et al. (2018) need \citep
- "The assumption that we have Σ" I think you are missing an "on" before "Σ".
- IIUC General X-regularization essentially enforces that the distance between a model trained on augmented + non-augmented data produces the same predictions as a model trained on only the non-augmented data when both models are fed additional unlabeled data. If this is the case I would suggest clarifying this a bit. This bears some similarity to consistency regularization/the "Pi-Model" (Laine & Aila 2017, Sajjadi et al. 2016), except in those cases consistency is enforced between a model fed with augmented and non-augmented versions of the same image. In your case IIUC you train two separate models and use unlabeled data to compare them; in the consistency regularization a single model is trained on labeled and unlabeled data together.
- "Hence empirically, our X-regularization has lower standard error but higher robust error than RSL." I don't think you have empirical evidence that supports this claim.

**Experience Assessment:**

I have published in this field for several years.

**Review Assessment: Checking Correctness Of Derivations And Theory:**

I assessed the sensibility of the derivations and theory.

**Review Assessment: Checking Correctness Of Experiments:**

I assessed the sensibility of the experiments.

**Review Assessment: Thoroughness In Paper Reading:**

I read the paper thoroughly.

---

> ### Author Response · Authors · 2019-11-15
> **Author Response to R1**
>
> We thank R1 for the feedback and their thoughtful summary. We have incorporated the suggestions for improvement in our revision. We request R1 to refer to the general comment above and evaluate the revised paper. In particular, we have clarified our setting of data augmentation and added more experiments and baselines for X-regularization.
>
> Regarding specific comments,
> - “Crosses unclear in figure”: Addressed in Fig 1
> - “Stairs word wrongly mentioned”: Fixed the typo in setting
> - “Examples to clarify setting”: Added examples of covariate-shifted data augmentation under setting
> - “Connection between linear regression setting and neural nets”: We added a new section, Section 5 in our revised paper that describes this clearly. As a quick summary, the theory suggests that as the size of the original dataset increases, the harmful effect of data augmentation decreases. We see this holds in practice.
> - “Emphasize that AT perturbations are label-preserving, and thus falls in our setting” We make this clear in the setting in the revision
> - “\citep missing”: fixed
> - “Assumption on \Sigma”: We make no assumption on \Sigma, the population covariance. Rather, we assume we have access to \Sigma via unlabeled data
> - “Comparison to semisupervised methods”: We added a brief comparison to standard semisupervised learning in Discussion (Section 7) and a more detailed comparison in Appendix H. To summarize, consistency regularization does not use pseudo-labels. X-regularization requires pseudo-labels from a model that is trained without augmentations
> - “X-reg vs. Robust Self Training (RST)”: Table 1 reports all relevant numbers. We would like to note that we realized that the methodological difference between what we proposed as X-reg and what was studied as RST in Carmon et al. 2019 is quite small. Hence, we revised the presentation to not claim X-reg as a new method, but rather view it as a generalization of RST. We view and present our contribution to be a theoretical justification for why RST improves standard accuracy.  Empirically, we additionally evaluate RST on more forms of adversarial training and perturbations such as rotations.

---

> > ### Comment · AnonReviewer1 · 2019-11-15
> > **Response**
> >
> > Thanks for your response.

---

### Official Review · AnonReviewer2 · 2019-10-28
**Official Blind Review #2**

**Rating:** 3

**Review:**

This paper describes situations whereby data augmentation (particularly drawn from a true distribution) can lead to increased generalization error even when the model being optimized is appropriately formulated.  The authors describe a somewhat intuitive example of how this can happen, and proceed to support this example and generalize with mathematical rigor.

The machinery of the analysis and the positive suggestion of X-regularization strike me as technically sophisticated and effective.

This is largely a theoretical paper.  The central example and description assume the availability of a true distribution to draw an arbitrary amount of examples from.  While CIFAR-10 is used to demonstrate the effectiveness of X-regularization, it is not compared to any other augmenatation / semi-supervised / regularization approach.  This makes contextualizing the contribution of this work more challenging.  General data augmentation techniques frequently do not draw examples form a true distribution (here the test distribution or a held-out development set) but rather manipulate real examples through some transformation function (like noise addition, rotation, scaling, warping, etc.).  This work does not address these conditions.  Rather the augmentation investigated here is more like self-training, where information from unlabeled examples are used in the training (i.e. regularization) of a model.  If i understand the technique correctly, assuming the eigenvector decomposition is relatively consistent under different samples of augmentation data, the regularization should be consistent whether regularized using the test set or some other unlabeled set.  While I think this is theoretically true, it would be useful to see this reflected on the CIFAR data, where the test data is unused during training at all.

The contextualization of this work to related work -- particularly self-training and semi-supervised training -- is quite thin.  While acknowledged, the technical rigor that is brought to the main result is not used to contrast with other techniques.  Neither are there empirical comparisons to existing techniques in this space.

On balance, I find the theoretical investigation and explanation for "how does this happen" to be more compelling than the presentation of X-regularization.

Note on "Experience Assessment": While I've done research on data augmentation and published in this space, the formulation and analytical tools employed in this paper I am less familiar with.

**Experience Assessment:**

I do not know much about this area.

**Review Assessment: Checking Correctness Of Derivations And Theory:**

I assessed the sensibility of the derivations and theory.

**Review Assessment: Checking Correctness Of Experiments:**

I assessed the sensibility of the experiments.

**Review Assessment: Thoroughness In Paper Reading:**

I read the paper at least twice and used my best judgement in assessing the paper.

---

> ### Author Response · Authors · 2019-11-15
> **Author Response to R2**
>
> We thank R2 for their feedback. The concerns raised stem from a confusion regarding the setting. R2 says that we assume that the augmented points are from the true distribution. We would like to clarify that we make *no* assumption on the augmented inputs. We only assume that the targets are from the true predictive/target distribution. As R2 points out, we typically augment with transformations. Since these transformations are label-preserving, our setting indeed covers these cases. We have revised the paper to make the setting more clear. We alco clarify that we differ from the standard semisupervised setting because we study both robustness (via augmentations) and standard accuracy. We request R2 to reconsider all our contributions with these clarifications. We have also expanded the discussions/contextualization of our work, experiments with more kinds of augmentation (rotations) and empirical comparisons to other relevant work. We request R2 to see the general comment above and also the revised paper.

---

### Author Response · Authors · 2019-11-15
**General Comment**

We thank the reviewers for their feedback. We have made substantial revisions to the paper that reflect the main concerns raised by the reviewers. The main changes are as follows:

- Modified the introduction: we now clearly state that our main motivation is to understand and prevent the well-documented drop in standard accuracy with adversarial augmentations via adversarial training.

- Elaborated on what we mean by “covariate-shifted” data augmentations: We allow arbitrary augmented inputs and only constrain their targets to come from the true target distribution. In particular, this includes augmenting with ‘label-preserving’ transformations such as small translations, flips, crops, etc. or imperceptible l-p perturbations.

- Contributions and positioning: The main contribution is to theoretically understand why training with augmented data can lead to an increase in standard error. This theoretical study motivates X-regularization, a general self-training-like procedure that leverages unlabeled data to fix the increase in test error. X-reg for robustness has been studied in recent work (Carmon et al. 2019, Najafi et al. 2019, Uesato et al. 2019) as Robust Self-Training (RST). Our work thus provides theoretical justification for why RST should also improve standard error. We note that standard semisupervised learning studies standard accuracy, while we study standard accuracy alongside robustness.

- Additional baselines and perturbation types: We added augmentations with random and adversarial rotations/translations. We additionally compare with standard semi-supervised self-training, Interpolated Adversarial Training and Neural Architecture Search. X-reg improves robust and standard error over vanilla counterparts, and the gains are comparable to IAT and NAS (Table 1).

---

### Decision · Program_Chairs · 2019-12-19

**Decision:**

Reject

**Comment:**

This paper describes situations whereby data augmentation (particularly drawn from a true distribution) can lead to increased generalization error even when the model being optimized is appropriately formulated. The authors propose "X-regularization" which requires that models trained on standard and augmented data produce similar predictions on unlabeled data. The paper includes a few experiments on a toy staircase regression problem as well as some ResNet experiments on CIFAR-10.  This paper received 2 recommendations for rejection, and one weak accept recommendation.  After the rebuttal phase, the author who recommended weak acceptance indicated their willingness to let the paper be rejected in light of the other reviews.  The reviewer highlighted: "I think the authors could still to better to relate their theory to practice, and expand on the discussion/presentation of X-regularization."  The main open issue is that the theoretical contributions of the paper are not sufficiently linked to the proposed algorithm.